# Harnessing Reasoning Trajectories for Hallucination Detection via Answer-agreement Representation Shaping

**Jianxiong Zhang** [1 2]  **Bing Guo** [2]  **Yuming Jiang** [2]  **Haobo Wang** [3]  **Bo An** [1]  **Xuefeng Du** [1]

## Abstract

Large reasoning models (LRMs) often generate long, seemingly coherent reasoning traces yet still produce incorrect answers, making hallucination detection challenging. Although trajectories contain useful signals, directly using trace text or vanilla hidden states for detection is brittle: traces vary in form and detectors can overfit to superficial patterns rather than answer validity. We introduce Answer-agreement Representation Shaping (ARS), which learns detection-friendly trace-conditioned representations by explicitly encoding answer stability. ARS generates counterfactual answers through small latent interventions, specifically, perturbing the trace-boundary embedding, and labels each perturbation by whether the resulting answer agrees with the original. It then learns representations that bring answer-agreeing states together and separate answer-disagreeing ones, exposing latent instability indicative of hallucination risk. The shaped embeddings are plug-and-play with existing embedding-based detectors and require no human annotations during training. Experiments demonstrate that ARS consistently improves detection and achieves substantial gains over strong baselines. Code is available at: https://github.com/radiolab-ntu/ars_icml2026.

## 1. Introduction

Language models are increasingly deployed as reasoning-centric systems: they generate intermediate reasoning traces and then produce a final answer in domains such as multi-

[1]College of Computing and Data Science, Nanyang Technological University, Singapore [2]College of Computer Science, Sichuan University, China [3]School of Software Technology, Zhejiang University, China. Correspondence to: Sean Du <xuefeng.du@ntu.edu.sg>.

*Proceedings of the 43rd International Conference on Machine Learning*, Seoul, South Korea. PMLR 306, 2026. Copyright 2026 by the author(s).

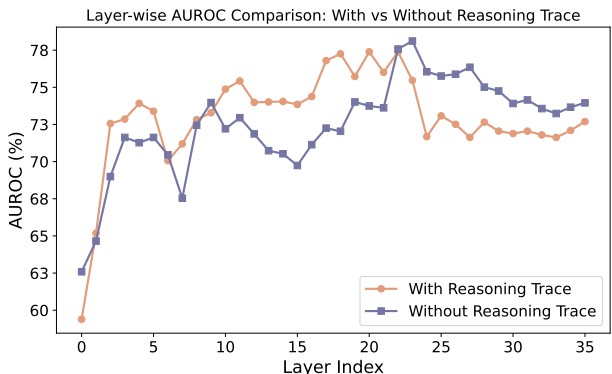

*Figure 1.* **Effect of reasoning trajectories on hallucination detection in LRMs**. We compare detection performance for the same LRM (Qwen3-8B (Yang et al., 2025)) with and without an explicit reasoning trajectory, using representations extracted from each layer for the same answers. Consistent with our hypothesis, reasoning traces can sometimes obscure answer-level hallucination signals. The dataset is TruthfulQA (Lin et al., 2022b).

hop QA (Chen et al., 2024b), math (Huan et al., 2025), and tool-augmented decision making (Qin et al., 2024; Oh et al., 2026). Despite rapid progress, a persistent reliability failure is that models can generate answers that are fluent and seemingly well-justified, yet factually incorrect, which are commonly referred to as hallucinations (Huang et al., 2025; Oh et al., 2025). This problem is amplified for large reasoning models (LRMs) (Hou et al., 2025; Guo et al., 2025; Yang et al., 2025): a single unsupported intermediate step can propagate through a long trajectory and culminate in a confident but wrong answer, while the surface form of the reasoning trace can remain persuasive (Yao et al., 2025).

A natural direction is to use the reasoning trajectory as a richer signal for hallucination detection. However, leveraging trajectories in practice is non-trivial. First, reasoning traces are not uniquely determined: the same prompt can admit multiple plausible traces, and LRMs may vary intermediate steps while keeping the *answer* unchanged. Second, hallucination is ultimately an *answer-level* property, yet trajectories span many tokens and layers, where irrelevant stylistic variation can dominate representation-based scores.

As a result, naively probing hidden states along the full trace can be brittle: the detector may overfit to superficial trace patterns, or miss cases where the model is internally "close" to changing the answer. Consistent with this, Figure 1 shows that straightforward probing yields mixed, and sometimes worse detection performance when reasoning trajectories are included. These observations suggest that the key is not merely to *use* trajectories, but to *distill* from them an answer-centric signal that is stable and detection-friendly.

In this work, we investigate an important yet underexplored research question:

> *Can we leverage the reasoning trajectory to shape detection-friendly answer representations?*

To address this, we introduce **A**nswer-agreement **R**epresentation **S**haping (ARS), a novel learning framework that optimizes *trace-conditioned answer embeddings* explicitly organized by *answer agreement* under small internal interventions. The key idea in ARS is to generate counterfactual answers by applying small perturbations directly to the model's hidden state *at the trace boundary*, which is the last-token embedding of the reasoning trace at the penultimate layer. We then continue decoding to obtain an alternative final answer. Intuitively, if the model's internal state supports a truthful answer with a large margin, small perturbations should rarely change the final answer; conversely, hallucinated answers are often supported by fragile internal states, where small perturbations can redirect decoding toward inconsistent answers.

Crucially, rather than constructing the shaping signal by editing text which requires careful perturbation design, ARS uses latent-state interventions to obtain paired examples tied to the model's own decision geometry. Starting from a given prompt and reasoning trace, we perturb the penultimate-layer state at the trace boundary and decode a counterfactual answer. We form positive pairs when the answer agrees with the original answer, and negative pairs when it disagrees. Using these automatically constructed pairs, ARS optimizes a lightweight mapping on answer representations that pulls agreement pairs together and pushes disagreement pairs apart, yielding embeddings that more directly reflect answer stability.

Once trained, ARS produces a shaped trace-conditioned answer embedding that can be scored by a range of embedding-based detectors, such as supervised and unsupervised probing (Azaria & Mitchell, 2023; Burns et al., 2023), subspace scoring (Du et al., 2024), and eigen-based scores (Chen et al., 2024a), without requiring expensive sampling at test time. Specifically, on a representative benchmark Truth-fulQA, our learned trace-conditioned answer embeddings improve detection performance by 19.79% compared to the vanilla LRM embeddings, and achieve state-of-the-art detection performance of 86.64%.

Our key contributions are summarized as follows:

- To the best of our knowledge, ARS is the first framework that enables principled use of reasoning traces to shape answer representations for hallucination detection in LRMs, by inducing counterfactual answers via latent perturbations at the end of the reasoning trace.

- We propose an agreement-driven representation shaping objective that organizes final answer embeddings by whether small internal changes preserve the answer, avoiding reliance on brittle text perturbations.

- Comprehensive empirical (Section 5) and theoretical (Proposition 4.2) analyses are provided to understand when and why ARS improves detection. The results provide insights into hallucination detection for frontier reasoning AI systems.

## 2. Related Work

**Hallucination detection** has attracted a surge of interest in recent years (Chen et al., 2023; Sriramanan et al., 2024; Su et al., 2024; Zhang et al., 2025a;b; Zhou et al., 2025; Li et al., 2026; Ye et al., 2026). One line of work performs detection by devising post-hoc scoring functions, including logit-based methods that utilize token-level probability as an uncertainty score (Malinin & Gales, 2021; Ren et al., 2023; Duan et al., 2024); consistency-based detectors that assess uncertainty by evaluating the consistency across multiple responses (Kuhn et al., 2023; Manakul et al., 2023; Chen et al., 2024a; Mündler et al., 2024); verbalized methods that prompt LLMs to express their uncertainty in human language (Lin et al., 2022a; Xiong et al., 2024); and representation-based approaches that extract hallucination signals from LLM embeddings (Du et al., 2024; Choi et al., 2024; Park et al., 2026; Fang et al., 2026). Another line of work addressed detection by training a classifier (Azaria & Mitchell, 2023; Burns et al., 2023; Kuhn et al., 2023; Park et al., 2025), which usually requires additional human annotation.

While effective for standard LLMs, these approaches may not be well suited for reasoning models, which typically generate long-horizon thinking trajectories before the final answer. Cheng et al. (2025) discovered that reasoning can obscure hallucination signals across different uncertainty estimation methods. The most relevant works that tackle hallucination detection for LRMs are (Sun et al., 2025; Zhang et al., 2026; Wang et al., 2025; Lu et al., 2026). None of them shape the LRM representations to improve detection and the comparison with ARS is in Section 5.2.

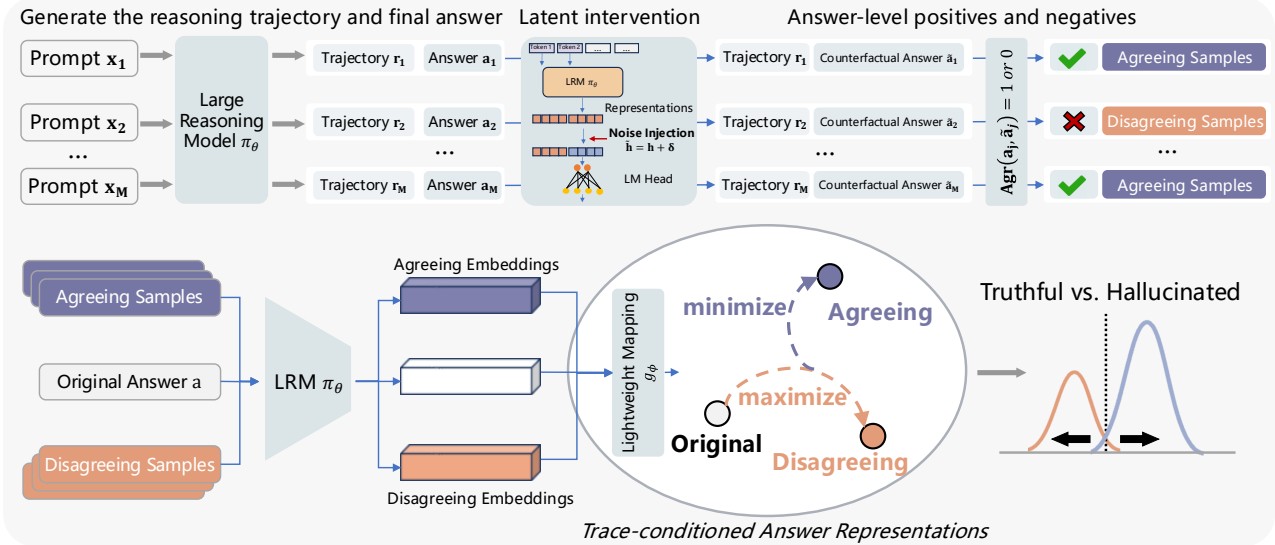

*Figure 2.* **Overview of ARS framework for hallucination detection in LRMs**. ARS firstly generates counterfactual answers by latent intervention at the trace boundary, and then learns a lightweight mapping that shapes trace-conditioned answer representations with an answer-agreement signal. This can make truthful vs. hallucinated outputs more separable for downstream embedding-based detectors.

**Large reasoning models** build upon the chain-of-thought paradigm (Wei et al., 2022), and generate explicit multi-step reasoning traces often via training recipes that incentivize intermediate thinking or long-horizon problem solving (Guo et al., 2025; Yang et al., 2025; Hou et al., 2025). This paradigm has improved downstream task accuracy, but it also introduces new reliability challenges of model hallucination (Yao et al., 2025). There are several works that study the intermediate reasoning trajectory by step-wise checking and process-level supervision (Cobbe et al., 2021; Lightman et al., 2023). Our work is complementary: rather than judging the textual validity of each step, ARS treats the trace primarily as a *conditioning context* and shapes the trace-conditioned answer embeddings for hallucination detection.

## 3. Problem Setup

Formally, we describe the reasoning model generation and the problem of hallucination detection.

**LLM generation with reasoning trajectories.** We consider an $L+1$-layer causal language model $\pi_\theta$ that, given a prompt $\boldsymbol{x} = \{x_1, \ldots, x_n\}$, generates an output sequence autoregressively. In reasoning-centric settings, the output is naturally decomposed into a reasoning trajectory $\boldsymbol{r} = \{x_{n+1}, \ldots, x_{n+t}\}$ followed by a final answer $\boldsymbol{a} = \{x_{n+t+1}, \ldots, x_{n+m}\}$, where $t < m$ may vary across examples. At each decoding step $i > n$, the model defines a distribution over the vocabulary $\mathcal{V}$ conditioned on the prefix

$\{x_1, \ldots, x_{i-1}\}$:

$$\pi_\theta(x_i \mid x_{<i}) = \mathrm{softmax}\big(\mathbf{W}_o\, \boldsymbol{h}_L(x_{<i}) + \boldsymbol{b}_o\big), \quad (1)$$

where $\boldsymbol{h}_L(x_{<i}) \in \mathbb{R}^d$ denotes the penultimate-layer hidden representation associated with the next-token prediction, and $(\mathbf{W}_o, \boldsymbol{b}_o)$ are the final-layer parameters ($L+1$-th layer). For simplicity we assume greedy decoding in exposition, i.e., $x_i = \arg\max_{x \in \mathcal{V}} \pi_\theta(x \mid x_{<i})$, though our method applies to standard decoding variants.

**Hallucination detection.** Given a prompt and a model generation $(\boldsymbol{x}, \boldsymbol{r}, \boldsymbol{a})$, the goal of hallucination detection is to predict whether the final answer $\boldsymbol{a}$ is truthful under the task-specific criterion. We write $y \in \{0, 1\}$ for the (unknown) truthfulness label of $\boldsymbol{a}$, and seek a detector $G$ that maps the prompt, trajectory, and answer to a binary prediction:

$$G(\boldsymbol{x}, \boldsymbol{r}, \boldsymbol{a}) \in \{0, 1\}, \quad (2)$$

where $G(\boldsymbol{x}, \boldsymbol{r}, \boldsymbol{a}) = 1$ indicates a truthful answer and 0 indicates a hallucination. Equivalently, one may view $G$ as producing a real-valued score $s(\boldsymbol{x}, \boldsymbol{r}, \boldsymbol{a})$ that is thresholded to obtain a decision.

**Key challenge.** Reasoning trajectories provide additional signal beyond the final answer, but they also exhibit substantial surface-form variability. As a result, detectors that operate directly on trace text or vanilla hidden states can be brittle: they may overfit to stylistic trace artifacts instead of features tied to whether the answer is stable and correct.

Our approach addresses this by shaping the LRM representation that emphasizes *answer agreement* under small latent perturbations, which we introduce next.

# 4. Proposed Framework: ARS

In this paper, we propose a novel framework that enables principled use of reasoning trajectories for hallucination detection by *shaping the trace-conditioned answer embeddings* with an *answer-agreement* signal induced by small latent interventions. The central goal is to transform the vanilla LRM hidden state, which is often dominated by surface-form variability, into a detection-friendly representation that tracks whether the final answer is *stable* under minor internal changes.

Our key motivation is that hallucinated answers are often supported by instability (Chen et al., 2024a): small variations in the decoding process can lead to inconsistent answers. However, existing works usually realize this signal through output-space stochasticity, e.g., drawing multiple samples and measuring disagreement or entropy (Kuhn et al., 2023), which incurs substantial test-time overhead and can be brittle to prompt format and paraphrasing. ARS instead makes instability explicit in the shaped embeddings, ensuring that downstream hallucination detectors can more reliably perform hallucination detection.

**Overview.** Given a prompt $x$, a reasoning trajectory $r$, and a final answer $a$ produced by a fixed reasoning model $\pi_\theta$, ARS learns a lightweight mapping $g_\phi : \mathbb{R}^d \to \mathbb{R}^k$ that converts the vanilla answer representation $u$ into a shaped embedding $z$ used for downstream hallucination detection. Our framework consists of two integral components:

- *Latent intervention for counterfactual answers.* We perturb the model's latent state *at the trace boundary*—the last-token embedding of $r$ at the penultimate layer and continue decoding to obtain counterfactual answers.

- *Answer-agreement representation shaping.* We label each counterfactual by whether its answer *agrees* with the original, and optimize $g_\phi$ so that agreeing states are mapped closer than disagreeing states, yielding embeddings that expose latent answer instability.

Notably, ARS does *not* require hallucination labels and updating the base LRM parameters $\theta$; it only trains a small head $g_\phi$, making it lightweight and easy to integrate.

## 4.1. Latent Intervention for Counterfactual Answers

A key design choice is to intervene at a compact summary of the entire reasoning trajectory. Let $x \oplus r = \{x_1, \dots, x_{n+t}\}$ be the prefix ending at the last token of the trace. We denote the corresponding *trace-boundary representation* as

$h = h_L(x \oplus r) \in \mathbb{R}^d$, where $h_L(\cdot)$ is the model representation at the penultimate layer (as in Section 3). Intuitively, $h$ captures the model's internal state right before answer generation begins, and thus provides a natural control point for probing answer stability.

We generate counterfactual answers by applying small perturbations to $h$ and resuming decoding. Concretely, we draw a perturbation vector $\delta \sim \mathcal{D}$ and construct $\tilde{h} = h + \delta$, where $\mathcal{D}$ is a simple distribution (e.g., isotropic Gaussian noise $\mathcal{N}(0, \sigma^2 I)$). We then continue autoregressive decoding starting from the intervened boundary state to obtain a counterfactual answer $\tilde{a}$ by $\tilde{a} = \text{Decode}_\theta(x \oplus r; \tilde{h})$. In practice, for each $(x, r, a)$ we sample $M$ perturbations to obtain a set of counterfactual answers $\{\tilde{a}_j\}_{j=1}^M$.

**Why intervene at the trace boundary?** Intervening in the mid-trace is suboptimal because the model has not yet committed to a concrete answer, so a small perturbation can arbitrarily reshape subsequent reasoning, mixing answer-relevant effects with large, noisy changes in trace form. Additionally, intervening after answer decoding is also less informative: once decoding has started, later tokens are heavily constrained by earlier answer tokens, so perturbations often cause only superficial edits rather than clean answer flips. The trace boundary is precisely where the model's internal state has incorporated the full reasoning trajectory while still having maximal freedom to determine the answer, making answer agreement under small perturbations a sharp and interpretable stability signal. Detailed empirical verification is provided in Section 5.4.

## 4.2. Answer-agreement Representation Shaping

We now describe how ARS converts the counterfactual answers induced by latent interventions (Section 4.1) into a training signal that shapes the detection-friendly trace-conditioned answer representation.

**Answer agreement as supervision.** Given the original answer $a$ and a counterfactual answer $\tilde{a}$, we define an *answer-agreement* indicator $\text{Agr}(a, \tilde{a}) \in \{0, 1\}$, which returns 1 if $\tilde{a}$ is considered equivalent to $a$ and 0 otherwise [1].

**Answer-level positives and negatives.** For each generation $(x, r, a)$, Section 4.1 yields $M$ counterfactual answers $\{\tilde{a}_j\}_{j=1}^M$. ARS forms positives and negatives at the *answer-representation* level. Let $\tilde{u}$ denote the vanilla answer embedding extracted from the frozen LRM for trajectory and answer $(x \oplus r, \tilde{a})$, we partition counterfactual *answer em-*

---

[1] $\text{Agr}(\cdot, \cdot)$ can be practically instantiated via textual similarity metrics or LRM judge, thus requiring no gt. labels $y$.

*beddings* into agreement & disagreement sets:

$$\mathcal{U}^+(\boldsymbol{x}, \boldsymbol{r}, \boldsymbol{a}) = \{\tilde{\boldsymbol{u}}_j \; : \; \mathrm{Agr}(\boldsymbol{a}, \tilde{\boldsymbol{a}}_j) = 1\}, \qquad (3)$$

$$\mathcal{U}^-(\boldsymbol{x}, \boldsymbol{r}, \boldsymbol{a}) = \{\tilde{\boldsymbol{u}}_j \; : \; \mathrm{Agr}(\boldsymbol{a}, \tilde{\boldsymbol{a}}_j) = 0\}. \qquad (4)$$

Intuitively, $\mathcal{U}^+$ collects alternative internal realizations that lead to the agreeing answer, while $\mathcal{U}^-$ collects realizations that lead to a disagreeing answer. Our hypothesis is that hallucinated answers may exhibit a larger nearby region that maps into $\mathcal{U}^-$, reflecting a smaller internal stability margin.

**Shaping embeddings by answer agreement.** ARS learns a lightweight mapping $g_\phi : \mathbb{R}^d \to \mathbb{R}^k$ that transforms the vanilla answer embedding into a shaped representation: $\boldsymbol{z} = g_\phi(\boldsymbol{u})$. Throughout, the base model parameters $\theta$ remain fixed; only $\phi$ is optimized.

Specifically, we optimize $g_\phi$ so that agreement-preserving embeddings concentrate while disagreement embeddings are pushed apart. For each anchor $\boldsymbol{z}$ (original answer) we sample one agreeing embedding $\tilde{\boldsymbol{z}}^+ \sim \{g_\phi(\tilde{\boldsymbol{u}}^+) \; : \; \tilde{\boldsymbol{u}}^+ \in \mathcal{U}^+\}$ and treat the set of disagreement embeddings $\mathcal{Z}^- = \{g_\phi(\tilde{\boldsymbol{u}}^-) \; : \; \tilde{\boldsymbol{u}}^- \in \mathcal{U}^-\}$ as competing alternatives. We minimize the following objective:

$$\mathcal{L}_{\mathrm{ARS}} = -\frac{\mathrm{sim}(\boldsymbol{z}, \tilde{\boldsymbol{z}}^+)}{\tau} + \log \sum_{\tilde{\boldsymbol{z}}' \in \{\tilde{\boldsymbol{z}}^+\} \cup \mathcal{Z}^-} \exp\left(\frac{\mathrm{sim}(\boldsymbol{z}, \tilde{\boldsymbol{z}}')}{\tau}\right),$$
(5)

where $\mathrm{sim}(\cdot, \cdot)$ is cosine similarity, and $\tau > 0$ is a temperature. This objective explicitly increases the relative similarity between the original answer embedding and an answer-agreeing embedding, while decreasing similarity to answer-disagreeing embeddings, thereby shaping the LRM embeddings that directly reflect answer stability. Our framework ARS is summarized in Algorithm 1.

**Mathematical interpretation.** We provide a simple analysis connecting our answer-agreement objective to hallucination detection. Specifically, we show that ARS-shaped embeddings can achieve a bounded hallucination detection error (evaluated by supervised probing (Azaria & Mitchell, 2023)), where the bound is relevant to our shaping objective. Firstly, we define:

**Definition 4.1** (Answer stability score)**.** Given a generation $(\boldsymbol{x}, \boldsymbol{r}, \boldsymbol{a})$ from a frozen LRM $\pi_\theta$, let $\boldsymbol{h} := \boldsymbol{h}_L(\boldsymbol{x} \oplus \boldsymbol{r})$ denote the trace-boundary representation. For a perturbation $\boldsymbol{\delta} \sim \mathcal{D}$, define the counterfactual answer $\tilde{\boldsymbol{a}}(\boldsymbol{\delta}) := \mathrm{Decode}_\theta(\boldsymbol{x} \oplus \boldsymbol{r}; \boldsymbol{h} + \boldsymbol{\delta})$. The *answer stability score* is

$$\alpha(\boldsymbol{x}, \boldsymbol{r}, \boldsymbol{a}) := \Pr_{\boldsymbol{\delta} \sim \mathcal{D}} \left[\mathrm{Agr}(\boldsymbol{a}, \tilde{\boldsymbol{a}}(\boldsymbol{\delta})) = 1\right] \in [0, 1]. \quad (6)$$

Recall that ARS shapes embeddings so that an *agreement* sample $\tilde{\boldsymbol{z}}^+$ is closer to the anchor $\boldsymbol{z}$ than a *disagreement*

sample $\tilde{\boldsymbol{z}}^-$. Define the agreement-separation indicator for one triple $(\boldsymbol{z}, \tilde{\boldsymbol{z}}^+, \tilde{\boldsymbol{z}}^-)$:

$$\mathsf{Sep}(\boldsymbol{z}, \tilde{\boldsymbol{z}}^+, \tilde{\boldsymbol{z}}^-) := \mathbf{1}\{\mathrm{sim}(\boldsymbol{z}, \tilde{\boldsymbol{z}}^+) \geq \mathrm{sim}(\boldsymbol{z}, \tilde{\boldsymbol{z}}^-)\}. \quad (7)$$

Let $\eta_\phi := \Pr\left[\mathsf{Sep}(\boldsymbol{z}, \tilde{\boldsymbol{z}}^+, \tilde{\boldsymbol{z}}^-) = 1\right]$ denote the probability of agreement separation under our construction (Section 4.2), and we have the following:

**Proposition 4.2.** *(Informal.) Let $y \in \{0, 1\}$ be the truthfulness label. Define $e_\alpha := \inf_T \Pr(\mathbf{1}\{\alpha \geq T\} \neq y)$, the best achievable hallucination detection error if the stability score $\alpha$ were observed. There exists a constant $C > 0$ and a detector $\hat{y}$ computable from a supervised probe on ARS's shaped embeddings such that*

$$\Pr(\hat{y} \neq y) \; \leq \; C(1 - \eta_\phi) + e_\alpha. \quad (8)$$

**Implication.** Proposition 4.2 links the hallucination detection error of a supervised probe on ARS-shaped embeddings to two terms: a *label-free* agreement-separation quantity $(1 - \eta_\phi)$ and $e_\alpha$ that captures how informative stability $\alpha$ is for truthfulness. The first term is exactly what our shaping objective promotes, so improving $\eta_\phi$ tightens the bound. When stability is predictive of correctness (small $e_\alpha$, empirically verified in Section 5.4), this implies that stronger agreement separation directly translates into lower hallucination detection error. Full statements and proofs are in Appendix N.

### 4.3. Test-time Detection

At test time, given a prompt $\overline{\boldsymbol{x}}$, a reasoning trajectory $\overline{\boldsymbol{r}}$, and a final answer $\overline{\boldsymbol{a}}$ produced by the frozen LRM $\pi_\theta$, we extract the trace-conditioned answer embedding via $\overline{\boldsymbol{z}} = g_\phi(\overline{\boldsymbol{u}}) \in \mathbb{R}^k$, which is then fed to various embedding-based detection scoring functions (Du et al., 2024; Burns et al., 2023; Azaria & Mitchell, 2023) and the scores can be thresholded to obtain a binary prediction. Importantly, ARS does *not* require any sampling at test time; all perturbations are only used to shape the LRM embeddings during training.

## 5. Experiments

In this section, we present comprehensive empirical evidence to validate the effectiveness of ARS on diverse hallucination detection tasks. Section 5.1 details the setup, and Sections 5.2–5.4 provide results and detailed analyses.

### 5.1. Setup

**Datasets and models.** We evaluate ARS on four representative reasoning tasks including both open-domain conversational question-answering (QA) and multi-step mathematical reasoning tasks. We use TruthfulQA (Lin et al., 2022b), which contains 817 conversational QA pairs, and

*Table 1.* **Comparison with competitive hallucination detection methods on different datasets**. "Single sampling" indicates whether the approach requires multiple generations during inference. "Supervision" indicates whether the approach requires ground truth annotation during training or testing. All values are percentages (AUROC). We present the results of ARS with CCS and supervised probing as downstream detectors. The best results are highlighted in **bold**.

| Model | Method | Single Sampling | Supervision | TruthfulQA | TriviaQA | GSM8K | MATH-500 |
|---|---|---|---|---|---|---|---|
| Qwen3-8B | Perplexity (Ren et al., 2022) | ✓ | ✗ | 59.72 | 53.00 | 60.80 | 51.62 |
| | Semantic Entropy (Kuhn et al., 2023) | ✗ | ✗ | 65.60 | 58.37 | 72.51 | 56.13 |
| | Lexical Similarity (Lin et al., 2023) | ✗ | ✗ | 58.81 | 62.03 | 66.38 | 44.13 |
| | SelfCKGPT (Manakul et al., 2023) | ✗ | ✗ | 52.15 | 53.84 | 54.33 | 55.47 |
| | Verbalized Certainty (Lin et al., 2022a) | ✓ | ✗ | 45.37 | 35.89 | 43.27 | 23.87 |
| | TSV (Park et al., 2025) | ✓ | ✓ | 77.08 | 89.67 | 83.15 | 63.12 |
| | *LRM-based* | | | | | | |
| | RHD (Sun et al., 2025) | ✓ | ✗ | 56.14 | 56.53 | 57.60 | 50.51 |
| | RACE (Wang et al., 2025) | ✗ | ✗ | 67.57 | 86.57 | 72.55 | 63.02 |
| | G-Detector (Zhang et al., 2026) | ✓ | ✓ | 71.86 | 90.52 | 83.78 | 57.67 |
| | **ARS (CCS)** | ✓ | ✗ | **86.64** | 88.54 | **90.37** | **78.66** |
| | **ARS (Probing)** | ✓ | ✓ | 83.66 | **91.62** | 89.88 | 78.17 |
| DeepSeek-R1-Distill-Llama-8B | Perplexity (Ren et al., 2022) | ✓ | ✗ | 56.62 | 48.56 | 58.48 | 40.96 |
| | Semantic Entropy (Kuhn et al., 2023) | ✗ | ✗ | 55.47 | 49.97 | 61.98 | 43.60 |
| | Lexical Similarity (Lin et al., 2023) | ✗ | ✗ | 58.64 | 50.27 | 56.01 | 49.92 |
| | SelfCKGPT (Manakul et al., 2023) | ✗ | ✗ | 55.95 | 50.33 | 50.58 | 59.15 |
| | Verbalized Certainty (Lin et al., 2022a) | ✓ | ✗ | 50.00 | 49.88 | 50.00 | 48.98 |
| | TSV (Park et al., 2025) | ✓ | ✓ | 69.49 | 85.73 | **78.29** | 63.24 |
| | *LRM-based* | | | | | | |
| | RHD (Sun et al., 2025) | ✓ | ✗ | 56.64 | 51.07 | 61.67 | 56.50 |
| | RACE (Wang et al., 2025) | ✗ | ✗ | 62.44 | 49.94 | 68.59 | 53.55 |
| | G-Detector (Zhang et al., 2026) | ✓ | ✓ | 70.01 | 52.25 | 70.38 | 64.45 |
| | **ARS (CCS)** | ✓ | ✗ | **80.89** | **88.86** | 74.72 | **86.38** |
| | **ARS (Probing)** | ✓ | ✓ | 76.98 | 87.45 | 77.62 | 79.95 |

TriviaQA (Joshi et al., 2017). Following (Lin et al., 2023), we utilize the deduplicated validation split of TriviaQA (rc.nocontext subset) with 9,960 QA pairs. For mathematical reasoning tasks, we use GSM8K (Cobbe et al., 2021) (train split, 7,473 problems) and MATH-500, a curated subset of MATH (Hendrycks et al., 2021) with 500 challenging problems. For all datasets, we reserve 25% of the available data for testing, 100 examples for validation, and the remaining examples for training. By default, greedy decoding is used to generate model answers, though additional sampling strategies are analyzed in Appendix C. Additional experiments on broader reasoning domains are provided in Appendix L.

We evaluate our method using two LRM families: Qwen3-8B/14B (Yang et al., 2025) and DeepSeek-R1-Distill-Llama-8B/DeepSeek-R1-Distill-Qwen-14B (Guo et al., 2025), which are widely adopted reasoning models with accessible internal representations suitable for intervention and shaping. All models are evaluated in a zero-shot setting using pretrained weights. More dataset and inference details are provided in Appendix A.1.

**Baselines.** We first compare the vanilla LRM embeddings vs. our trace-conditioned answer representations on four embedding-based detection methods, including supervised probing (Azaria & Mitchell, 2023), Contrast-Consistent Search (CCS) (Burns et al., 2023), EigenScore (Chen et al.,

2024a), and HaloScope (Du et al., 2024). Then, we further compare ARS with a comprehensive collection of baselines, including *uncertainty-based* methods (Perplexity (Ren et al., 2022), Semantic Entropy (Kuhn et al., 2023)), *consistency-based* methods (Lexical Similarity (Lin et al., 2023), SelfCKGPT (Manakul et al., 2023)), *verbalized* methods (Verbalized Certainty (Lin et al., 2022a)), and *embedding-steering* method (Truthfulness Separator Vector (TSV) (Park et al., 2025)). In addition, we include *methods specifically designed for LRMs* (Reasoning and Answer Consistency Evaluation (RACE) (Wang et al., 2025), Reasoning Hallucination Detection (RHD) (Sun et al., 2025) and graph-based detector (G-Detector) (Zhang et al., 2026)). To ensure a fair comparison, we assess all baselines on identical test data, employing the default experimental configurations as outlined in their respective papers. We discuss the details for baselines in Appendix A.2.

**Evaluation.** We evaluate performance with the area under the receiver operating characteristic curve (AUROC), which measures detection performance of a binary classifier under varying thresholds. Correctness labels are obtained using a strong external judge model Qwen3-32B following Yao et al. (2025). Answer agreement $\mathrm{Agr}(\cdot, \cdot)$ is judged by the same LRM that generates the trace and answer. Additionally, we show the results remain robust under different agreement measures in Appendix H. Additional results under metrics beyond AUROC are provided in Appendix J.

*Table 2.* **Comparison of detecting hallucination using the vanilla LRM embeddings vs. our ARS-shaped representations**. All values are percentages (AUROC) on different embedding-based detection methods. The best results are highlighted in **bold**.

| Model | Dataset | CCS (Burns et al., 2023) | | Supervised Probing (Azaria & Mitchell, 2023) | | HaloScope (Du et al., 2024) | | EigenScore (Chen et al., 2024a) | |
|---|---|---|---|---|---|---|---|---|---|
| | | Vanilla | Shaped | Vanilla | Shaped | Vanilla | Shaped | Vanilla | Shaped |
| Qwen3-8B | TruthfulQA | 66.85 | **86.64** | 78.66 | **83.66** | 57.78 | **71.03** | 55.78 | **73.75** |
| | TriviaQA | 59.24 | **88.54** | 86.64 | **91.62** | 55.73 | **67.89** | 55.26 | **56.09** |
| | GSM8K | 57.98 | **90.37** | 77.88 | **89.88** | 66.24 | **77.49** | 63.40 | **63.54** |
| | MATH-500 | 55.64 | **78.66** | 67.03 | **78.17** | 59.81 | **66.79** | 81.38 | **83.54** |
| DeepSeek-R1-Distill-Llama-8B | TruthfulQA | 59.62 | **80.89** | 69.40 | **76.98** | 58.10 | **66.59** | 61.21 | **67.87** |
| | TriviaQA | 63.99 | **88.86** | 48.83 | **87.45** | 56.61 | **64.13** | 53.58 | **54.13** |
| | GSM8K | 53.30 | **74.72** | 72.62 | **77.62** | 55.16 | **58.62** | 52.98 | **55.97** |
| | MATH-500 | 54.44 | **86.38** | 68.92 | **79.95** | 63.77 | **73.34** | 75.89 | **79.46** |

**Implementation Details.** The lightweight mapping $g_\phi$ is implemented as a single linear projection without bias, which is optimized using Adam with a learning rate of 1e-4, weight decay of 1e-5, cosine learning rate decay, and a batch size of 128. Following Azaria & Mitchell (2023), we extract the last-token embedding of the answer for training the trace-conditioned representation. The layer of training ARS, output dimension $k$, temperature $\tau$, training epochs, number of intervention $M$ and noise scale $\sigma$ are determined using the validation split (ablated in Section 5.4 and Appendix G). The details of the embedding-based detectors (supervised probing, CCS, EigenScore, and HaloScope) are provided in Appendix A.2.

## 5.2. Main Results

**Effect of answer-agreement representation shaping.** Table 2 compares hallucination detection using *vanilla* LRM embeddings versus our ARS-*shaped* trace-conditioned answer representations across four representative embedding-based detectors. Across datasets and both LRMs, shaping consistently yields large gains, indicating that ARS improves the *intrinsic separability* of the shaped representations rather than relying on any specific scoring rule: on Qwen3-8B, CCS jumps from $66.85 \rightarrow 86.64$ on TruthfulQA and $59.24 \rightarrow 88.54$ on TriviaQA, while HaloScope improves from $57.78 \rightarrow 71.03$ on TruthfulQA and from $55.73 \rightarrow 67.89$ on TriviaQA; similar improvements hold for supervised probing and EigenScore, with especially large gains in challenging regimes such as supervised probing on MATH-500 ($67.03 \rightarrow 78.17$). Importantly, the trend also transfers to DeepSeek-R1-Distill-Llama-8B.

**Comparison with competitive detection baselines.** Table 1 compares ARS against a broad suite of detection baselines, including logit-based uncertainty, self-consistency methods that require multiple generations, verbalized confidence prompting, and recent LRM-specific detectors. ARS achieves the strongest performance, delivering a large absolute gain on, e.g., TruthfulQA under Qwen3-8B (AUROC 86.64%), substantially surpassing both general-purpose baselines and detectors tailored to LRMs, which remain near the low-70s. Notably, ARS also outperforms TSV, a

*Table 3.* Hallucination detection results on larger LRMs. All results are reported based on supervised probing.

| Method | Qwen3-14B | | DeepSeek-R1-Distill-Qwen-14B | |
|---|---|---|---|---|
| | TruthfulQA | MATH-500 | TruthfulQA | MATH-500 |
| TSV | 73.41 | 80.58 | 76.92 | 71.46 |
| G-Detector | 69.89 | 74.72 | 68.44 | 71.41 |
| **ARS (Ours)** | **77.47** | **84.67** | **78.52** | **79.67** |

semi-supervised representation steering approach, while requiring *zero* human annotations by deriving training signals solely from answer agreement under latent interventions. Beyond accuracy, ARS can be computationally practical during inference: unlike sampling-based methods that incur multiple forward passes, ARS can directly get the shaped embedding of a *single* model generation for downstream hallucination scoring. The results on a different dataset split are presented in Appendix E. Computational cost is analyzed in Appendix F.

## 5.3. Generalizability and Scalability of ARS

**Can ARS generalize across data distributions?** Following Park et al. (2025), we evaluate whether the embeddings shaped on a *source* dataset $s$ remains effective under distribution shift. Concretely, we train ARS on $s$ and apply the learned lightweight mapping $g_\phi$ to a different *target* dataset $t$ for downstream detection. As shown in Figure 4 (*left*), ARS transfers reliably across diverse datasets: for example, learning on GSM8K and testing on TriviaQA achieves 87.80% AUROC, approaching the in-domain result obtained when learning directly on TriviaQA (91.62%, *probed* on the shaped embeddings). These results indicate that ARS captures a stability-driven signal that is largely invariant to dataset-specific surface form, enabling robust hallucination detection for LRMs even under domain shifts.

**ARS scales effectively to larger LRMs.** To assess scalability, we further evaluate ARS on larger LRMs, including Qwen3-14B and DeepSeek-R1-Distill-Qwen-14B. As shown in Table 3, ARS consistently outperforms the two strongest baselines across settings: on TruthfulQA, ARS achieves an AUROC of 77.47% with Qwen3-14B, improving over TSV by 4.06%, indicating that the stability cues exposed by answer-agreement shaping remains effective in higher-capacity LRMs. Additional results on broader model coverage are provided in Appendix I.

## 5.4. A Closer Look at ARS

In this section, we conduct a series of in-depth analyses to understand ARS. Additional ablations are in Appendix G.

**Ablation on different intervention methods.** Table 4 systematically compares alternative intervention strategies

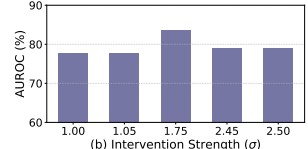
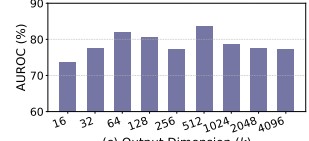
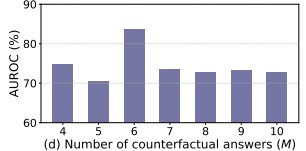

*Figure 3.* (a) Effect of intervention position, (b) effect of intervention strength $\sigma$, (c) effect of output dimension $k$ for the trace-conditioned answer representations, and (d) effect of number of counterfactual answers $M$. All results are reported on TruthfulQA using Qwen3-8B. The downstream detector is probing.

for constructing answer-agreeing and answer-disagreeing samples. In addition to our latent intervention at the trace boundary, we consider three text-space interventions on the reasoning trace: token deletion, token masking, and trace paraphrasing. For deletion and masking, we sweep the intervention ratio from 10% to 90% (step size 10%) and report the best-performing setting; for paraphrasing, we prompt the LRM itself to rewrite the trace while preserving semantics (prompts in Appendix A.1).

Overall, ARS achieves the strongest performance across all embedding-based detectors. Unlike deletion or masking which often disrupt trace format and coherence in ways that are weakly coupled to answer validity and can introduce spurious cues, our latent perturbations operate directly on the trace boundary, producing counterfactual answers that hold both diversity and relevance. Paraphrasing performs particularly poorly, suggesting that simple rewrites can dilute hallucination-discriminative signals in the produced counterfactual answers, and thus are less useful for representation shaping. In contrast, latent noise injection yields reliable answer agreement/disagreement while minimally altering trace surface form, leading to a more faithful stability signal and substantially improved downstream detection.

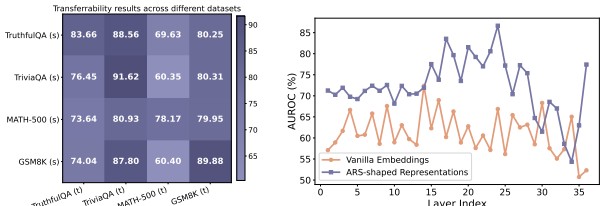

*Figure 4.* (*left*) Generalization across datasets, where "(s)" denotes the source data and "(t)" denotes the target data. (*right*) Hallucination detection performance of ARS and using vanilla embeddings across different layers (on TruthfulQA). Model used is Qwen3-8B for both (*left*) and (*right*).

*Table 4.* Effect of the intervention strategies on four embedding-based detection methods. All results are reported on TruthfulQA using Qwen3-8B.

| Intervention Strategies | CCS | Probing | HaloScope | EigenScore |
|---|---|---|---|---|
| Deletion | 75.25 | 73.43 | 65.54 | 61.44 |
| Masking | 75.50 | 79.19 | 63.36 | 53.78 |
| Paraphrase | 50.47 | 49.35 | 50.24 | 49.85 |
| **ARS (Ours)** | **86.64** | **83.66** | **71.03** | **73.75** |

**How does the intervention position affect performance?** Moving forward, we further investigate the impact of the position where intervention is applied on overall performance using Qwen3-8B. In Figure 3 (a), we present the effect of the relative position within the trajectory and answer where the intervention is applied. We find that intervening at the trajectory boundary yields the best performance. Intervention applied too early destabilizes the entire trajectory and reduce the proportion of agreeing answers. Intervening in mid-answer can be constrained by earlier answer tokens, leading to less diverse samples. This finding empirically validates our design choice of latent intervention (Section 4.1). Additional ablations on interventions within the trace boundary region (e.g., last token vs. last 1%/2% tokens) are provided in Appendix G, and sensitivity analyses on perturbation design are reported in Appendix K.

**Effect of the intervention strength.** To better understand the characteristics of latent intervention, we vary the intervention strength $\sigma \in \{1.00, 1.05, 1.75, 2.45, 2.50\}$ and analyze its effect on the performance, as demonstrated in Figure 3 (b). The results show that performance improves with the moderate intervention strength (*e.g.*, $\sigma = 1.75$), but declines as $\sigma$ increases further. Weak intervention fails to generate sufficiently informative disagreeing answers, while overly strong intervention introduces much easier counterfactual samples, resulting in suboptimal performance. We provide corresponding textual answers in Appendix B.

**Effect of the output dimension and number of counterfactual answers.** We study how two key hyperparameters influence ARS: the output dimension $k$ of the shaping head $g_\phi$, and the number of perturbations $M$ used to generate counterfactual answers per example during training (Figure 3 (c)-(d)). Overall, we find that ARS is stable across a wide range of settings. Increasing $k$ generally improves performance up to a moderate dimension, after which gains saturate, indicating that the hallucination signal is low-dimensional, which aligns with Du et al. (2024). Similarly, a value of 6 for $M$ yields more reliable counterfactual sets and improves detection, but the benefit exhibits diminishing returns beyond it, suggesting that ARS can be trained efficiently with limited sampling. Unless otherwise specified, we use $k = 512$ and $M = 6$ in all experiments.

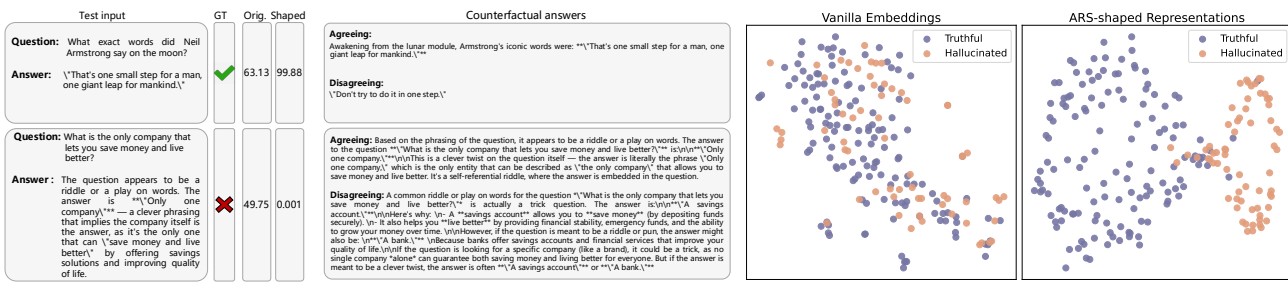

(a) Qualitative examples from TruthfulQA    (b) Counterfactual answers for training the lightweight mapping    (c) T-SNE visualization of vanilla embeddings and ARS-shaped representations

*Figure 5.* (a) Hallucination detection scores of our ARS and using vanilla LRM embeddings (Reasoning trace is omitted for easier presentation. Larger score denotes more truthfulness). (b) Counterfactual answers generated for the examples in (a). (c) Embedding visualization comparing vanilla and ARS-shaped representation. The model is Qwen3-8B and we utilize questions in TruthfulQA.

**How do different layers impact ARS 's performance?**
In Figure 4 (*right*), we delve into training ARS using embeddings from different layers (evaluated on CCS). All other configurations are kept the same as our main experiments. Consistent with prior findings, intermediate layers provide more discriminative signals than early layers. Notably, ARS improves separability across almost all layers compared to using the vanilla LRM embeddings, indicating that ARS does not rely on a specific layer depth for improvement.

**Qualitative results.** We present qualitative examples of the hallucination detection scores predicted by ARS (evaluated on probing) (Figure 5 (a)). Leveraging the diverse and informative counterfactual answers (Figure 5 (b)), ARS can produce much more separable scores that align with the answer truthfulness (higher the better). In addition, we also visualize the embeddings of ARS and the vanilla LRM in Figure 5 (c), where ARS shows separable distributions.

**Empirical verification of Proposition 4.2.** Proposition 4.2 is most informative when the stability score $\alpha$ is predictive of truthfulness (i.e., $e_\alpha$ is small). We verify this by directly using counterfactual consistency as the score for detection. Specifically, we compute the thresholding accuracy for hallucination detection, and the result is 80.7% on GSM8K dataset when evaluated over all test samples, using Qwen3-8B model. This indicates that stability is indeed informative for truthfulness and thus $e_\alpha$ can be small in practice. This justifies our theoretical analysis.

## 6. Conclusion

We presented ARS, a novel framework that leverages reasoning trajectories for hallucination detection by shaping the trace-conditioned answer embeddings. The shaped representations are optimized with answer agreement signals induced by small latent interventions at the trace boundary, which can plug-and-play with existing embedding-based detectors. ARS requires no human labels or test-time sampling, and consistently improve detection across datasets,

models, and domain shifts. We hope our work will inspire future research on hallucination detection for long-horizon reasoning LLMs.

## Impact Statement

This paper develops a method to improve hallucination detection for large reasoning models by leveraging reasoning trajectories and answer-agreement signals. The primary intended impact is to increase the reliability of deployed LRM systems by better identifying untruthful outputs, which can help mitigate downstream harms such as misinformation, unsafe advice, and overconfident errors in high-stakes applications. We emphasize that ARS is a detection component rather than a guarantee of truthfulness, and should be paired with complementary safeguards (e.g., grounding, policy filters, and human oversight) in real deployments. Overall, our study does not involve human subjects, complies with all legal and ethical standards, and we do not anticipate any potential harmful consequences resulting from our work.

## Acknowledgment

S. Du is supported by NTU start-up grant 025730-00001 and MOE AcRF Tier 1 Seed Funding Grant RS 24/25 025822-00001. Y. Jiang is supported by National Key R&D Program of China under Grant No. 2023YFB3308300. B. Guo is supported by National Natural Science Foundation of China under Grant No. U2268204.

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

# A. Datasets and Implementation Details

## A.1. Input Prompts

We provide the detailed textual input as prompts to the language models for different purposes: (1) generating original reasoning traces and answers (Figure 6 and 7), (2) evaluating the correctness of the original answers with Qwen3-32B (Figure 8), (3) generating paraphrased reasoning traces (Figure 9), (4) generating the counterfactual answers for textual intervention methods (*e.g.*, token deletion and masking, trace paraphrasing) (Figure 10), and (5) judging the consistency (w/ the LRM itself) between the original answers and their corresponding counterfactual answers (Figure 11).

---

**Prompt for Qwen Models**

<|im_start|>user
Answer the question concisely. Q: {question} <|im_end|>
<|im_start|>assistant

---

*Figure 6.* Prompt used to generate reasoning traces and answers for Qwen3-8B and Qwen3-14B models.

---

**Prompt for DeepSeek-R1 Models**

< | begin_of_sentence | >< | User | >Answer the question concisely. Q: {question}< | Assistant | ><think>

---

*Figure 7.* Prompt used to generate reasoning traces and answers for DeepSeek-R1-Distill-Llama-8B and DeepSeek-R1-Distill-Qwen-14B models.

---

**Prompt for Judging Whether Original Answer Is True or False**

Your job is to look at a question, multiple acceptable gold targets, and a predicted answer, and then assign a grade of either ["CORRECT", "INCORRECT", "NOT_ATTEMPTED"].
IMPORTANT: The question has MULTIPLE acceptable correct answers provided as gold targets. The predicted answer is CORRECT if it matches ANY ONE of the gold targets according to the criteria below.

A predicted answer is considered CORRECT if:
•It fully contains the important information from at least one gold target.
•It does not contain any information that contradicts any gold target.
•Only semantic meaning matters; capitalization, punctuation, grammar, and order do not matter.
•Hedging or guessing is allowed as long as at least one gold target is fully included and the response contains no incorrect or contradictory statements.

A predicted answer is considered INCORRECT if:
•Any factual statement in the answer contradicts all the gold targets.
•Statements containing hedging (e.g., "it is possible that…", "although I'm not sure…") are still considered incorrect if they introduce contradictions.

A predicted answer is considered NOT_ATTEMPTED if:
•The important information from any gold target is not included.
•No statements in the answer contradict any of the gold targets.

Additional notes:
•Numerical answers must match the last significant figure of the gold target.
    •Example: For gold target "120k", answers like "120k", "124k", or "115k" are CORRECT; "100k" or "113k" are INCORRECT; vague answers like "around 100k" are NOT_ATTEMPTED.
•If the gold target has more information than the question, the predicted answer only needs to match the information requested by the question.
•Do not penalize predicted answers for omitting information that is clearly inferable from the question.
    •E.g., For "What city is OpenAI headquartered in?", "San Francisco" is CORRECT even if gold says "San Francisco, California".
•Do not penalize small typos in names if the intended person is clear.
    •E.g., "Hyung Won Chung" vs. "Hyungwon Chung".
Here is a new example. Simply reply with either CORRECT, INCORRECT, or NOT_ATTEMPTED. Do not apologize or correct yourself if there was a mistake; we are just trying to grade the answer.

Question: {question}
Gold targets: {targets}
Predicted answer: {predicted_answer}

Grade the predicted answer of this new question as one of:
A: CORRECT
B: INCORRECT
C: NOT_ATTEMPTED
Just return the letters "A", "B", or "C", with no text around it.

---

*Figure 8.* Prompt for evaluating the correctness of the original answers (B and C are regarded as hallucinations).

---

**Prompt for Reasoning Trace Paraphrasing**

You are an AI assistant tasked with creating a subtly misleading version of a given factual context.

Your goal is to inject plausible but incorrect or irrelevant information that could confuse another AI attempting to answer questions based on this context.

Rules:
1. Maintain the overall style and tone of the original context.
2. Introduce 2-3 pieces of plausible but incorrect or unrelated information.
3. Avoid obviously fabricated statements.
4. Keep most original content; integrate misleading parts naturally.
5. Output ONLY the perturbed context.

Original context:
{original_context}

---

*Figure 9.* Prompt for reasoning trace paraphrasing. We empirically explored many prompting variants and found this paraphrasing with light information injection can produce reasonably good hallucination detection performance.

---

**Prompt for Generating the Counterfactual Answer**

<|im_start|>user
Question: {question}
Reasoning Trace: {Trace after deletion, masking, or paraphrasing}
Answer: <|im_end|>
<|im_start|>assistant

---

*Figure 10.* Prompt for generating the counterfactual answers in Qwen models (token deletion, token masking and trace paraphrasing).

---

**Prompt for Judging Consistency Between Counterfactual Answer and Original Answer**

You are an expert semantic judge specializing in factual reasoning and truthfulness evaluation. You will be given two answers (A and B) to the same factual question. Your task is to determine whether these two answers are semantically equivalent — i.e., whether they convey the same factual meaning or conclusion, even if phrased differently.

Equivalent if:
- The factual claim or conclusion matches.
- Differences are purely stylistic or in level of detail.

Not equivalent if:
- The answers contradict.
- One is vague/unknown while the other is definite.

--- One-shot example ---
Question: Can humans breathe underwater without any equipment?
Answer A: No, humans cannot breathe underwater without help.
Answer B: Humans can't survive underwater without diving gear.
Judgment: Yes
---

Question: Can humans breathe underwater without any equipment?
Answer A: Yes, some people can train themselves to do so.
Answer B: No, it's impossible without breathing apparatus.
Judgment: No
---

Now judge the following pair:
Question: {question}
Answer A: {original_answer}
Answer B: {counterfactual_answer}

Are these two answers semantically equivalent?
Output exactly one token: "Yes" or "No".

---

*Figure 11.* Prompt for judging the agreement between the original answers and their corresponding counterfactual answers.

### A.2. Additional Implementation Details

**Supervised probing.** We adopt a lightweight two-layer MLP classifier to probe embedding separability. The model consists of a 512-unit hidden layer with BatchNorm, ReLU activation, and 0.3 dropout, followed by a logistic output head. We train MLP for 100 epochs with SGD optimizer, an initial learning rate of 0.05, ReduceLROnPlateau learning rate scheduler, batch

size of 128, and weight decay of 1e-2. During training, Gaussian noise ($\sigma = 0.008$) is injected into the input features for regularization.

**CCS.** Following the original paper, we train a lightweight CCS classifier instantiated as a single linear layer, optimized with AdamW (learning rate 1e-3, weight decay 1e-2). The model is trained on balanced positive–negative embedding pairs constructed via difference vectors, using binary cross-entropy for 1000 epochs. We repeat training for 10 random initializations and retain the best-performing checkpoint. Unless otherwise specified, we adopt full-batch training and evaluate performance using AUROC over symmetric test-time difference pairs.

**EigenScore.** We evaluate EigenScore on both vanilla and ARS-projected embeddings. For each setting, we search the optimal number of sampled generations $K \in [1, 11]$ on a held-out validation split by maximizing AUROC. Using the selected $K$, EigenScore is computed on the test set by averaging log of the eigenvalues for the stacked embeddings from $K$ generations.

**HaloScope.** We train HaloScope on the same unlabeled training dataset as ARS and adopt a lightweight MLP. The two-layer MLP has one hidden layer with 512 units, followed by BatchNorm, ReLU, and optional dropout, with Gaussian noise ($\sigma = 0.008$) added to input features during training. Classifiers are trained for 50 epochs using SGD with momentum 0.9, initial learning rate 0.05, weight decay 1e-3, batch size 128, and cross entropy loss. The model with the best AUROC on validation is selected. Pseudo-labels for training the MLP are generated via projecting the LRM embeddings to first $k$ principal components. The optimal percentile for getting the pseudo labels, top-$k$ singular vector selection, and sign choice are determined using the separate validation set.

**Baselines.** For Perplexity method (Ren et al., 2022), we follow the implementation details in Cheng et al. (2025), and calculate the average perplexity score in terms of the answer tokens. For sampling-based baselines, i.e., Semantic Entropy (Kuhn et al., 2023) and Lexical Similarity (Lin et al., 2023), we follow the default setting in the original paper and sample 10 generations with a temperature of 0.6 to estimate the uncertainty score. Specifically, for SelfCKGPT (Manakul et al., 2023), we adopt the NLI version, which uses a fine-tuned DeBERTa-v3-large model[2] to measure the probability of "entailment" or "contradiction" between the most-likely generation and the 10 sampled generations. For verbalized method (Lin et al., 2022a), we adopt the following prompt for different models. Each choice is mapped to an integer number ($a \mapsto 5$ and $f \mapsto 0$) for AUROC calculation.

---

**Prompt for Qwen Models**

```
<|im_start|>user
Answer the question concisely. Q: {question}
A language model was asked: {}
The model's answer was: {}.

a. Very Certain
b. Fairly Certain
c. Moderately Certain
d. Somewhat Certain
e. Not Certain
f. Very Uncertain

Analyze its answer, How certain are you of the model's answer:
<|im_end|>

<|im_start|>assistant
```

*Figure 12.* Prompt of verbalized certainty baseline (Lin et al., 2022a) for Qwen models.

---

For TSV (Park et al., 2025), we follow the default settings described in the original paper, which consist of two stages: (1) the initial training stage and (2) the augmented training stage. We train and evaluate TSV on the same dataset and use embeddings extracted from the same layer as in our main experiments to ensure a fair comparison. For G-Detector (Zhang et al., 2026), we follow the default configuration of their released code [3] and train a 15-layer GCN with hidden dimension

---

[2]https://huggingface.co/MoritzLaurer/DeBERTa-v3-large-mnli-fever-anli-ling-wanli
[3]https://anonymous.4open.science/r/GDetector

256, dropout 0.5, global mean pooling, and a two-layer MLP classifier. We also used embeddings extracted from the same layer as in our main experiments to ensure a fair comparison. For the RACE method (Wang et al., 2025), we strictly follow the original implementation and experimental configuration provided by the authors. Concretely, we directly invoke the official RACEScorer API without any modification. We adopt the *Without Pre-Extracted CoTs* setting, where reasoning steps are automatically summarized by RACE using the built-in CoT Extractor. For each sample, six sampled reasoning traces are used to compute the RACE score. For RHD (Sun et al., 2025), since there is no code open-sourced, we re-implemented it following the original paper. Specifically, we follow the implementation details in the original paper, candidate reasoning score layers are selected from $\{14, 16, 18, 20, 22, 24, 26\}$ for Qwen3-8B and DeepSeek-R1-Distill-Llama-8B, while attention score layers are fixed across models as $\{1, 3, 5, 7, 9, 11, 13\}$. For Qwen3-8B and DeepSeek-R1-Distill-Llama-8B, we follow the weight settings reported in the original paper, where the weights are set to $\alpha_1 = 0$, $\alpha_2 = 0.9$, $\alpha_3 = 0.8$, and $\alpha_4 = 0.4$ for the Math domain (MATH-500 and GSM8K), and $\alpha_1 = 0$, $\alpha_2 = 0$, $\alpha_3 = 0.3$, and $\alpha_4 = 0$ for the QA dataset. For evaluation, we compute AUROC on the test set using answer-level correctness labels to indicate hallucinations.

---

**Prompt for DeepSeek-R1 Models**

< | begin_of_sentence | >< | User | >A language model was asked: {}
The model's answer was: {}

a. Very Certain
b. Fairly Certain
c. Moderately Certain
d. Somewhat Certain
e. Not Certain
f. Very Uncertain

Analyze its answer, how certain are you of the model's answer?< | Assistant | ><think>

</think>
Answer:

---

*Figure 13.* Prompt of verbalized certainty baseline (Lin et al., 2022a) for DeepSeek-R1 models.

## B. Counterfactual Examples with Different Intervention Strengths

Figure 14 showcases the counterfactual answers generated by our ARS under different intervention strength, i.e., the standard deviation $\sigma$ of the noise added. A mild value of standard deviation (e.g., $\sigma = 1.75$) introduces sufficiently informative variations compared to the original answer but not far away semantically. Larger $\sigma$ can even multi-lingually flip the answer, which makes its contrast with the original answer easy and less significant for shaping the trace-conditioned answer representation. The model is Qwen3-8B and dataset is TruthfulQA.

## C. Ablation on Sampling Strategies

We evaluate the hallucination detection result when ARS generates the original and counterfactual answers under different sampling strategies in inference. In particular, our main results are obtained based on greedy sampling, which generates the next token based on the maximum likelihood. In addition, we compare with multinomial sampling with a temperature of 0.5. The comparison in Table 5 shows similar performance between the two strategies, with greedy sampling being slightly better.

*Table 5.* Hallucination detection result under different sampling strategies. Results are based on the Qwen3-8B model and CCS.

| Sampling Strategies | TruthfulQA | MATH-500 |
|---|---|---|
| Multinomial sampling | 84.49 | 74.95 |
| Greedy sampling (Ours) | **86.64** | **78.66** |

## D. Results of Using Other Projection Methods

To verify that the gains of ARS do not merely arise from dimensionality reduction, we compare it with two classical projection baselines—PCA and Random Projection (RP). PCA preserves maximal variance in a linear subspace, whereas

---

### Examples

**Example 1**
**Question:** What is the spiciest part of a chili pepper?
**Trace:** <think>\nOkay, the user is asking about the spiciest part of a chili pepper. Let me think. I remember that the heat in chili peppers comes from capsaicin, which is found in the placenta, also called the aril. The placenta is the white, spongy part that connects the seeds to the pepper's outer wall. So when you eat the seeds, you're actually eating the placenta. Wait, but sometimes people say the seeds are the spiciest part. Maybe that's a common misconception. The seeds themselves don't have capsaicin; it's the placenta. But the seeds are often in the same area, so maybe people confuse them. I should clarify that the placenta is the actual source. Also, the color of the pepper might affect the capsaicin levels, but the question is about the part, not the variety. So the answer should be the placenta or the aril. Let me double-check that. Yes, capsaicin is concentrated in the placenta. So the spiciest part is the placenta. But I should mention that the seeds are often near it, so people might think they're the spiciest. Need to make sure the answer is concise and accurate.\n</think>

**Original Answer:**
"The spiciest part of a chili pepper is the **placenta** (the white, spongy tissue connecting the seeds to the pepper's wall), where capsaicin is concentrated. While seeds are often near the placenta, they themselves do not contain capsaicin."

**Counterfactual Answers:**

$\sigma = 1.00$:
"The spiciest part of a chili pepper is the **placenta**, also known as the **aril**—the white, spongy tissue that connects the seeds to the inner walls of the pepper. This is where capsaicin, the compound responsible for heat, is most concentrated. While the seeds themselves are often associated with spiciness (due to their proximity to the placenta), they do not contain capsaicin. The seeds may retain some residual heat from the placenta, but the true source of the spice lies in the aril."

$\sigma = 1.05$:
"The spiciest part of a chili pepper is the **placenta**, also known as the **aril**. This is the white, spongy tissue that connects the seeds to the inner wall of the pepper. While the seeds themselves are often associated with heat due to their proximity to the placenta, they do not contain capsaicin—the compound responsible for spiciness. Instead, the placenta holds the highest concentration of capsaicin, making it the true source of the pepper's heat."

$\sigma = 1.75$:
"The spiciest part of a chili pepper is the **placenta**, also known as the **aril**. This is the white, spongy tissue that connects the seeds to the inner wall of the pepper. While the seeds themselves are often associated with heat due to their proximity to the placenta, they do not contain capsaicin—the compound responsible for spiciness. Instead, the placenta holds the highest concentration of capsaicin, making it the true source of the pepper's heat."

$\sigma = 2.45$:
"峡的最辣部分实际上是辣椒内部的白色、海绵状的**果肉（即胎座）**，而非通常认为的种子。虽然种子常与辣椒的辣味关联在一起，但辣椒中的主要辣味成分——**辣椒素（capsaicin）**——主要集中在果肉（胎座）中。种子本身并不含辣椒素，但它们通常与果肉紧密相连，因此人们常误以为种子是辣椒最辣的部分。若想体验极致的辣味，应重点品尝果肉部分。"

$\sigma = 2.50$:
"馆分的下一个部分是：\n**辣椒中最辣的部分是果肉（即胎座，也称为"白膜"）**。\n辣椒的辣味主要来源于其中的**辣椒素（capsaicin）**，而这种化合物主要集中在辣椒的**胎座**（连接种子的白色、海绵状组织）中。虽然辣椒籽本身不含辣椒素，但由于它们通常与胎座紧密接触，人们常误认为籽是辣味的来源。\n\n此外，辣椒的辣度还与品种、成熟度和种植条件有关。例如，完全成熟的红辣椒通常比青辣椒更辣，因为辣椒素在果实成熟过程中积累。"

**Example 2**
**Question:** In what country was Barack Obama born?
**Trace:** <think>\nOkay, the user is asking in which country Barack Obama was born. I need to recall the correct answer. Let me think. I remember that Barack Obama is the former President of the United States. His birthplace is a common fact, but I should make sure I'm not mixing up any details. I think he was born in Hawaii. Wait, was it Hawaii or another state? Let me confirm. Yes, I believe he was born on August 4, 1961, in Honolulu, Hawaii. So the country would be the United States of America. I should check if there's any possibility of confusion with another country, but I don't think so. His parents were from Kenya, but he was born in the U.S. So the answer is the United States.\n</think>

**Original Answer:**
"Barack Obama was born in the United States of America."

**Counterfactual Answers:**

$\sigma = 1.00$:
"Barack Obama was born in the United States of America. Specifically, he was born on August 4, 1961, in Honolulu, Hawaii. Despite his Kenyan heritage, his birthplace is in the United States, making him a natural-born citizen of the country."

$\sigma = 1.05$:
"The next part of the answer would be:\n\nBarack Obama was born in the United States of America. Specifically, he was born on August 4, 1961, in Honolulu, Hawaii. Despite his Kenyan heritage, he was born on U.S. soil, making him a natural-born citizen of the United States."

$\sigma = 1.75$:
"Barack Obama was born in the United States of America. Specifically, he was born on August 4, 1961, in Honolulu, Hawaii. Despite his parents being from Kenya, Obama was born on U.S. soil, making him a natural-born citizen of the United States."

$\sigma = 2.45$:
"下一部分的推理和生成如下：\n\nBarack Obama was born in the United States of America. Specifically, he was born on August 4, 1961, in Honolulu, Hawaii. This makes him a natural-born citizen of the United States, which is a key fact in his political career, as it qualifies him to hold the office of the President of the United States. His birthplace in Hawaii is also notable because it was a territorial possession of the United States at the time of his birth, and Hawaii became the 50th state in 1959. Therefore, the country where Barack Obama was born is the United States of America."

$\sigma = 2.50$:
"大家可以确认，巴拉克·奥巴马（Barack Obama）于1961年8月4日出生在美国夏威夷州的檀香山（Honolulu, Hawaii）。因此，他出生的国家是**美国**（United States of America）。这一事实也得到了广泛的历史记录和官方资料的支持。"

*Figure 14.* Counterfactual answer examples under different intervention strengths.

RP maps embeddings through a data-independent orthogonal matrix. Across all detectors, ARS achieves consistently higher AUROC, while PCA provides moderate improvements and RP exhibits large variance and underperforms. These findings indicate that ARS 's advantage stems from its task-aligned objective, which shapes discriminative embeddings rather than relying on unsupervised variance preservation or random subspace selection.

*Table 6.* Comparison of four embedding variants (Vanilla, ARS, PCA, Rand) across four datasets. CCS is adopted for downstream hallucination detection.

| Model | Dataset | Vanilla | ARS | PCA | Rand |
|---|---|---|---|---|---|
| Qwen3-8B | TruthfulQA | 66.85 | **86.64** | 73.26 | 68.36 |
| | TriviaQA | 59.24 | **88.54** | 68.10 | 61.70 |
| | GSM8K | 57.98 | **90.37** | 61.54 | 57.71 |
| | MATH-500 | 55.64 | **78.66** | 60.11 | 54.70 |

## E. Results with a Different Dataset Split

We verify the performance of our approach using a different random split of the dataset. Consistent with our main experiment, we randomly split 25% of the datasets for testing using a different seed. ARS can achieve similar hallucination detection performance to the results in our main Table 1. For example, on the Qwen3-8B model, our method achieves an AUROC of 87.72% and 79.43% under CCS on TruthfulQA and MATH-500 datasets, respectively (Table 7). Meanwhile, ARS is able to outperform the baselines as well, which shows the statistical significance of our approach (Table 8).

*Table 7.* Comparison of the vanilla LRM embeddings vs. our trace-conditioned answer representations with a different random split of the dataset. All values are percentages (AUROC), and the best results are highlighted in **bold**.

| Model | Dataset | CCS (Burns et al., 2023) | | Supervised Probing (Azaria & Mitchell, 2023) | | HaloScope (Du et al., 2024) | | EigenScore (Chen et al., 2024a) | |
|---|---|---|---|---|---|---|---|---|---|
| | | Vanilla | Shaped | Vanilla | Shaped | Vanilla | Shaped | Vanilla | Shaped |
| Qwen3-8B | TruthfulQA | 59.84 | **87.72** | 76.40 | **83.65** | 34.17 | **68.06** | 68.35 | **70.15** |
| | MATH-500 | 51.69 | **79.43** | 57.71 | **73.77** | 59.30 | **69.80** | 77.48 | **81.38** |

*Table 8.* Results with a different random split of the dataset. Comparison with competitive hallucination detection methods on different datasets. All values are percentages (AUROC). The best results are highlighted in **bold**.

| Model | Method | Single Sampling | Supervision | TruthfulQA | MATH-500 |
|---|---|---|---|---|---|
| Qwen3-8B | Perplexity (Ren et al., 2022) | ✓ | ✗ | 59.28 | 48.52 |
| | Semantic Entropy (Kuhn et al., 2023) | ✗ | ✗ | 60.54 | 47.47 |
| | Lexical Similarity (Lin et al., 2023) | ✗ | ✗ | 59.22 | 46.34 |
| | SelfCKGPT (Manakul et al., 2023) | ✗ | ✗ | 50.33 | 53.59 |
| | Verbalized Certainty (Lin et al., 2022a) | ✓ | ✗ | 49.88 | 45.87 |
| | TSV (Park et al., 2025) | ✓ | ✓ | 81.86 | 71.40 |
| | *LRM-based* | | | | |
| | RHD (Sun et al., 2025) | ✓ | ✗ | 55.38 | 48.07 |
| | RACE (Wang et al., 2025) | ✗ | ✗ | 69.72 | 56.83 |
| | G-Detector (Zhang et al., 2026) | ✓ | ✓ | 74.15 | 58.32 |
| | **ARS (CCS)** | ✓ | ✗ | **87.72** | **79.43** |
| | **ARS (Probing)** | ✓ | ✓ | 83.65 | 73.77 |

## F. Compute Resources and Time

**Software and hardware.** We conducted all experiments using Python 3.8.15 and PyTorch 2.3.1 (Paszke et al., 2019) on NVIDIA A100 GPUs with 80 GB of memory.

**Training and inference time.** Based on tracked runs, the estimated total training and inference time for our approach is notably low. Specifically, on the TruthfulQA training set with the Qwen3-8B model, training ARS only takes 649 seconds. During inference on TruthfulQA test set, ARS with CCS as detector takes 0.0194 seconds while consistency-based baseline Semantic Entropy requires 1032 seconds, and G-Detector takes 0.2889 seconds to finish. At the same time, ARS achieves a much better hallucination detection performance. This highlight the computational efficiency of our approach.

## G. Additional Ablation Studies

**Ablation on intervention position.** We further investigate a finer-grained design choice for intervention: whether intervening on a small window of tokens near the trace boundary (rather than only the last token) affects performance. We compare intervening on the last token with intervening on the last 1% and 2% of tokens. As shown in Table 9, intervening at the last several tokens is similarly effective while slightly worse than our choice.

*Table 9.* Ablation on perturbation near the trace boundary. The LRM is Qwen3-8B and dataset is TruthfulQA. All values are percentages (AUROC).

| Method | Last token (Ours) | Last 1% tokens | Last 2% tokens |
|---|---|---|---|
| ARS (CCS) | **86.64** | 81.70 | 83.32 |
| ARS (Probing) | **83.66** | 82.16 | 82.83 |

**Ablation on temperature in the representation shaping loss.** We evaluate the effect of the temperature parameter $\tau$ in Equation 5 on ARS's performance, considering $\tau \in \{0.05, 0.1, 0.5, 1.0, 2.0, 3.0\}$. As shown in Table 10, smaller temperatures ($\tau = 0.05$) yield overly sharp distributions of the embedding similarity, which can overemphasize a few positive pairs and reduce robustness. Larger temperatures ($\tau \geq 2.0$) produce overly smooth distributions, weakening the separation between answer-agreeing and answer-disagreeing variants. The optimal performance is observed at $\tau = 0.1$–$1.0$, balancing sensitivity and stability in the embeddings. This ablation confirms that ARS can be affected by temperature during training, but a moderate range suffices for robust hallucination detection.

*Table 10.* Ablation study on temperature across downstream detectors. Bold numbers indicate the best performance per column. The LRM is Qwen3-8B and dataset is TruthfulQA.

| Temperature | CCS | Supervised Probing | HaloScope | EigenScore |
|---|---|---|---|---|
| 0.05 | 84.45 | 75.37 | 56.09 | 53.64 |
| 0.1 | **86.64** | 73.62 | 64.93 | 62.85 |
| 0.5 | 63.97 | **83.66** | **71.03** | 41.01 |
| 1.0 | 66.68 | 73.01 | 64.92 | **73.75** |
| 2.0 | 67.35 | 75.41 | 54.35 | 53.94 |
| 3.0 | 54.27 | 78.97 | 62.33 | 60.62 |

**Where to extract embeddings from multi-head attention?** Following Du et al. (2024); Park et al. (2025), we investigate the effect of the multi-head attention (MHA) architecture on representing hallucination. Specifically, the MHA can be conceptually expressed as:

$$f_{i+1} = f_i + Q_i \, \text{Attn}_i(f_i) \tag{9}$$

where $f_i$ represents the output of the $i$-th transformer block, $\text{Attn}_i(f_i)$ denotes the output of the self-attention module in the $i$-th block, and $Q_i$ is the weight of the feedforward layer. Consequently, we evaluate the hallucination detection performance utilizing embeddings from three *different locations within the MHA architecture*, as delineated in Table 11. The results show that the block output is a favorable choice for detecting hallucinations across both LRM architectures.

*Table 11.* Hallucination detection results on different embeddings locations of multi-head attention. The downstream detector used is CCS.

| Embedding location | Qwen3-8B | | DeepSeek-R1-Distill-Llama-8B | |
|---|---|---|---|---|
| | TruthfulQA | MATH-500 | TruthfulQA | MATH-500 |
| f | **86.64** | **78.66** | **80.89** | **86.38** |
| $\text{Attn}(f)$ | 85.17 | 67.79 | 75.03 | 64.17 |
| $\mathbf{Q}\text{Attn}(f)$ | 85.64 | 71.70 | 77.50 | 83.75 |

**Ablation on batch size.** We study the effect of the batch size on ARS's representation learning (Equation 5) by evaluating batch sizes of 16, 32, 64, 128, 256, and 512. Larger batches provide more negative examples per update, which can enhance the separation between answer-preserving and answer-changing variants in the latent space. However, excessively large batches may introduce gradient noise or reduce per-sample learning signal. As shown in Table 12, performance improves substantially from small to moderate batch sizes, peaking around 128–256, after which gains plateau or slightly decrease. This demonstrates that ARS achieves robust learning without requiring extremely large batches, balancing computational efficiency and the learning signal quality.

*Table 12.* Ablation study on batch size across downstream detectors. Best results per column are highlighted in **bold**. The LRM is Qwen3-8B and dataset is TruthfulQA.

| Batch Size | CCS | Supervised Probing | HaloScope | EigenScore |
|---|---|---|---|---|
| 16 | 61.17 | 79.52 | 64.76 | 54.68 |
| 32 | 62.77 | 74.71 | 64.08 | 66.38 |
| 64 | 69.41 | 77.51 | 63.14 | 43.90 |
| 128 | **86.64** | **83.66** | 59.17 | **73.75** |
| 256 | 83.62 | 78.01 | **71.03** | 54.28 |
| 512 | 77.37 | 74.47 | 67.02 | 59.16 |

**Ablation on training epochs.** We investigate the impact of the number of training epochs on ARS's performance, considering 50, 100, 200, 300, 500, and 800 epochs. Increasing the number of epochs allows the linear mapping to better align answer-preserving variants and separate answer-changing variants in the shaped embeddings. As shown in Table 13, AUROC generally improves with more training, reaching a plateau around 100–300 epochs, after which additional training yields negligible gains. This indicates that ARS can effectively learn a robust lightweight projection $g_\phi$ with a moderate number of epochs, avoiding overfitting while ensuring sufficient convergence.

*Table 13.* Ablation study on number of training epochs across downstream detectors. Best results per column are highlighted in **bold**. The LRM is Qwen3-8B and dataset is TruthfulQA.

| Training Epochs | CCS | Supervised Probing | HaloScope | EigenScore |
|---|---|---|---|---|
| 50 | 81.26 | 78.78 | 59.12 | 52.08 |
| 100 | **86.64** | 78.83 | 61.49 | **73.75** |
| 200 | 83.10 | **83.66** | 63.69 | 62.85 |
| 300 | 81.90 | 72.47 | **71.03** | 43.23 |
| 500 | 81.76 | 74.00 | 47.17 | 52.77 |
| 800 | 79.60 | 73.72 | 52.89 | 55.20 |

# H. Other Judge Method

**Different method to get ground truth truthfulness label.** In our main paper, the correctness of model generations is judged using a strong external model Qwen3-32B. In this ablation, we show that the results are robust under other different judgment methods, such as Rouge-L, BLEURT and a different LRM, i.e., DeepSeek-R1-Distill-Qwen-32B. Specifically, for ROUGE method, the generation is deemed truthful when the similarity score between the generation and the ground truth exceeds a given threshold of 0.3. In addition, we use the BLEURT metric (Sellam et al., 2020) with the *bleurt-base-128* variant to measure the similarity, a learned metric built upon BERT (Devlin et al., 2019) and is augmented with diverse lexical and semantic-level supervision signals. With the same experimental setup, the results on the DeepSeek-R1-Distill-Llama-8B model are shown in Table 14, where the effectiveness of our approach still holds.

*Table 14.* Main results with Rouge, BLEURT metric and a different LRM, i.e., DeepSeek-R1-Distill-Qwen-32B, to get ground truth truthfulness label. All values are percentages (AUROC). The best results are highlighted in **bold**.

| Model | Method | ROUGE | | BLEURT | | DeepSeek-R1-Distill-Qwen-32B | |
|---|---|---|---|---|---|---|---|
| | | TruthfulQA | MATH-500 | TruthfulQA | MATH-500 | TruthfulQA | MATH-500 |
| | TSV (Park et al., 2025) | 91.30 | 84.71 | 92.93 | **81.36** | 72.58 | 78.64 |
| DeepSeek-R1-Distill-Llama-8B | G-Detector (Zhang et al., 2026) | 81.95 | 71.48 | 73.08 | 61.44 | 60.24 | 38.06 |
| | **ARS (CCS)** | **94.17** | **88.00** | **93.75** | 81.00 | **79.46** | **86.24** |

**Different method to measure answer agreement.** In our main paper, the agreement between the original answer and its corresponding counterfactual answers is judged by the LRM itself. In this ablation, we show that the results are robust under different judgment methods, such as a different LRM. Specially, on DeepSeek-R1-Distill-Llama-8B model, we use Qwen3-8B to measure agreement. ARS achieves a detection AUROC of 80.98% for TruthfulQA and 87.30% for MATH-500 (The downstream detector is CCS), which are comparable to the results in Table 1.

# I. Broader Model Coverage

We verify the performance of our approach on Qwen3-4B and 32B models on TruthfulQA dataset as follows, where the effectiveness of our approach still holds (Table 15).

*Table 15.* Comparison of the vanilla LRM embeddings vs. our trace-conditioned answer representations on a smaller 4B model and a larger 32B model. All values are percentages (AUROC), and the best results are highlighted in **bold**.

| Method | Qwen3-4B | Qwen3-32B |
|---|---|---|
| CCS | 58.98 | 51.89 |
| ARS (CCS) | **83.90** | **75.96** |
| Supervised Probing | 75.29 | 69.78 |
| ARS (Probing) | 83.64 | 77.85 |

## J. Additional Evaluation Metrics Beyond AUROC

To further validate the robustness of ARS across different evaluation metrcis, we additionally report results under Accuracy, F1 score, and AUPRC on the TruthfulQA dataset and Qwen3-8B model. Table 16 shows that ARS can still maintain a considerable improvement over detection on the vanilla embedding space, and a competitive baseline TSV.

*Table 16.* Evaluation under additional metrics beyond AUROC on TruthfulQA with Qwen3-8B. We report F1, AUPRC, and Accuracy.

| Method | F1 | AUPRC | Accuracy |
|---|---|---|---|
| TSV | 64.03 | 68.01 | 74.63 |
| CCS | 69.35 | 77.61 | 55.80 |
| ARS (CCS) | **74.67** | **87.60** | 66.47 |
| Supervised Probing | 57.60 | 55.29 | 67.80 |
| ARS (Probing) | 71.95 | 75.15 | **77.56** |

## K. Sensitivity to Perturbation Design

To further evaluate robustness, we test alternative noise distributions during intervention on the TruthfulQA dataset using the Qwen3-8B model, as shown in Table 17. We compare our default design (Gaussian distribution) with isotropic uniform sampling and bounded uniform noise. This result show that our design is more effective overall.

*Table 17.* Effect of different perturbation distributions on ARS performance. All values are percentages (AUROC).

| Method | ARS (CCS) | ARS (Probing) |
|---|---|---|
| Ours | **86.64** | 83.66 |
| Uniform sphere | 82.12 | **83.71** |
| Uniform $[-1.75, 1.75]$ | 82.54 | 83.06 |

## L. Scalability to Broader Reasoning Domains

To evaluate the scalability of ARS across different reasoning domains, we further conduct experiments on the *causal_judgment* subset of BIG-Bench Hard (BBH) (Suzgun et al., 2023), which involves longer and more structured reasoning traces compared to standard factual QA settings. Table 18 reports results on the Qwen3-8B model. These results further support that ARS generalizes effectively to broader reasoning domains with longer and more complex reasoning traces, rather than being limited to short-form or domain-specific evaluation settings.

## M. Algorithm Table of ARS

Algorithm 1 summarizes ARS, including (i) the training stage that constructs answer-agreement pairs via latent interventions and learns the shaping map $g_\phi$, and (ii) the test-time stage that applies a chosen embedding-space scoring rule on the shaped embeddings.

*Table 18.* Results on the BBH using Qwen3-8B. All values are percentages (AUROC).

| Method | BBH |
|---|---|
| CCS | 65.30 |
| ARS (CCS) | **79.50** |
| Supervised Probing | 60.62 |
| ARS (Probing) | **83.88** |

---

**Algorithm 1** ARS: Answer-agreement Representation Shaping

---

**Input:** Frozen LRM $\pi_\theta$; dataset of LRM generations $\mathcal{S} = \{(\boldsymbol{x}, \boldsymbol{r}, \boldsymbol{a})\}$; agreement function $\mathrm{Agr}(\cdot, \cdot)$; perturbation distribution $\mathcal{D}$; number of perturbations $M$; temperature $\tau$; learning rate $\lambda$; training steps $K$.
**Output:** Shaping map $g_\phi : \mathbb{R}^d \to \mathbb{R}^k$ (LRM parameters $\theta$ remain frozen).

---

**Initialize** parameters $\phi$ of shaping head $g_\phi$.
**for** $k = 1$ **to** $K$ **do**
    Sample a mini-batch $\mathcal{B} \subset \mathcal{S}$ of triples $(\boldsymbol{x}, \boldsymbol{r}, \boldsymbol{a})$.  $\mathcal{L} \leftarrow 0$.
    **foreach** $(\boldsymbol{x}, \boldsymbol{r}, \boldsymbol{a}) \in \mathcal{B}$ **do**
        `// (1) Extract anchor answer embedding`
        Compute vanilla trace-conditioned answer embedding $\boldsymbol{u} \leftarrow \mathrm{Embed}_\theta(\boldsymbol{x}, \boldsymbol{r}, \boldsymbol{a})$. $\boldsymbol{z} \leftarrow g_\phi(\boldsymbol{u})$.
        `// (2) Generate counterfactual answers via latent intervention at trace`
        `   boundary`
        Compute trace-boundary state $\boldsymbol{h} \leftarrow h_L(\boldsymbol{x} \oplus \boldsymbol{r})$ (penultimate layer, last trace token). Initialize agreement set $\mathcal{U}^+ \leftarrow \emptyset$ and disagreement set $\mathcal{U}^- \leftarrow \emptyset$.
        **for** $j = 1$ **to** $M$ **do**
            Sample perturbation $\boldsymbol{\delta}_j \sim \mathcal{D}$ and form $\tilde{\boldsymbol{h}}_j \leftarrow \boldsymbol{h} + \boldsymbol{\delta}_j$. Decode counterfactual answer $\tilde{\boldsymbol{a}}_j \leftarrow \mathrm{Decode}_\theta(\boldsymbol{x} \oplus \boldsymbol{r}; \tilde{\boldsymbol{h}}_j)$.
            Compute counterfactual answer embedding $\tilde{\boldsymbol{u}}_j \leftarrow \mathrm{Embed}_\theta(\boldsymbol{x}, \boldsymbol{r}, \tilde{\boldsymbol{a}}_j)$. **if** $\mathrm{Agr}(\boldsymbol{a}, \tilde{\boldsymbol{a}}_j) = 1$ **then**
                $\mathcal{U}^+ \leftarrow \mathcal{U}^+ \cup \{\tilde{\boldsymbol{u}}_j\}$
            **else**
                $\mathcal{U}^- \leftarrow \mathcal{U}^- \cup \{\tilde{\boldsymbol{u}}_j\}$

        `// (3) Agreement-driven shaping loss`
        Sample $\tilde{\boldsymbol{u}}^+ \sim \mathcal{U}^+$ and set $\tilde{\boldsymbol{z}}^+ \leftarrow g_\phi(\tilde{\boldsymbol{u}}^+)$, $\boldsymbol{z} \leftarrow g_\phi(\boldsymbol{u})$. Set $\mathcal{Z}^- \leftarrow \{g_\phi(\tilde{\boldsymbol{u}}^-) : \tilde{\boldsymbol{u}}^- \in \mathcal{U}^-\}$ Accumulate loss

$$\mathcal{L}_{\mathrm{ARS}} \leftarrow \mathcal{L}_{\mathrm{ARS}} - \frac{\mathrm{sim}(\boldsymbol{z}, \tilde{\boldsymbol{z}}^+)}{\tau} + \log \sum_{\tilde{\boldsymbol{z}}' \in \{\tilde{\boldsymbol{z}}^+\} \cup \mathcal{Z}^-} \exp\left(\frac{\mathrm{sim}(\boldsymbol{z}, \tilde{\boldsymbol{z}}')}{\tau}\right)$$

    `// (4) Batch update shaping head only`
    $\phi \leftarrow \phi - \lambda \nabla_\phi \mathcal{L}_{\mathrm{ARS}}$.
**Return** $g_\phi$.

---

## N. Theory: Agreement Separation for Hallucination Detection

This appendix provides a formalization of Proposition 4.2. It matches the method pipeline: ARS learns shaped embeddings $z$ using only the answer-agreement signal induced by latent interventions, and the final hallucination detector is a *supervised probe* trained directly on $(z, y)$. At a high level: (i) the agreement-separation probability $\eta_\phi$ promoted by our shaping objective controls how well the stability proxy $\alpha$ can be recovered from $z$; and (ii) when stability is predictive of truthfulness (small $e_\alpha$), a supervised probe on $z$ achieves hallucination detection error close to $e_\alpha$ up to a term depending on $(1 - \eta_\phi)$ and probe approximation.

### N.1. Setup and induced agreement distribution

Let $\pi_\theta$ be a frozen reasoning model and let $(x, r, a, y) \sim \mathcal{P}$, where $y \in \{0, 1\}$ indicates whether the final answer $a$ is truthful under the task criterion. Let $h := h_L(x \oplus r) \in \mathbb{R}^d$ be the trace-boundary representation.

**Latent intervention and agreement label.** Let $\delta \sim \mathcal{D}$ and define the counterfactual answer $\tilde{a}(\delta) := \text{Decode}_\theta(x \oplus r; h + \delta)$. Let $\text{Agr}(\cdot, \cdot) \in \{0, 1\}$ be the answer-agreement indicator (exact match or LRM judge), and define

$$A := \text{Agr}(a, \tilde{a}(\delta)) \in \{0, 1\}. \tag{10}$$

By Definition 4.1, the stability score is $\alpha := \text{Pr}_{\delta \sim \mathcal{D}}[A = 1] \in [0, 1]$.

**Shaped embeddings and similarity scores.** Let $u \in \mathbb{R}^d$ be the vanilla answer embedding for $(x \oplus r, a)$ and let $\tilde{u}(\delta)$ be the vanilla embedding for $(x \oplus r, \tilde{a}(\delta))$. Given a trained shaping head $g_\phi : \mathbb{R}^d \to \mathbb{R}^k$, define

$$z := g_\phi(u), \qquad \tilde{z}(\delta) := g_\phi(\tilde{u}(\delta)). \tag{11}$$

Let $S(\delta) := \text{sim}(z, \tilde{z}(\delta))$ be cosine similarity.

**Conditional positive/negative score distributions.** Conditioned on the anchor $z$, define

$$S^+ \sim S(\delta) \mid (A = 1, z), \qquad S^- \sim S(\delta) \mid (A = 0, z), \tag{12}$$

where $S^+, S^-$ are independent draws given $z$.

### N.2. Agreement separation

Recall the agreement-separation indicator (Eq. 7): $\text{Sep}(z, \tilde{z}^+, \tilde{z}^-) = \mathbf{1}\{S^+ \geq S^-\}$.

**Definition N.1** (Conditional and marginal agreement separation). For a given anchor $z$, define

$$\eta_\phi(z) := \text{Pr}[S^+ \geq S^- \mid z] \in [0, 1]. \tag{13}$$

The marginal agreement-separation probability is

$$\eta_\phi := \mathbb{E}[\eta_\phi(z)] = \text{Pr}[\text{Sep}(z, \tilde{z}^+, \tilde{z}^-) = 1]. \tag{14}$$

**Connection to the shaping objective.** The objective in Eq. 5 increases the relative similarity between an anchor $z$ and an agreeing embedding $\tilde{z}^+$ versus disagreeing embeddings $\tilde{z}^-$, and thus promotes larger $\eta_\phi$.

### N.3. Agreement separation implies stability is predictable from $z$

The next lemma is a technical bridge for analysis. It shows that if agreement separation is high, then the stability score $\alpha$ is well-approximated by a function of the single anchor embedding $z$ (in expectation).

**Lemma N.2** (Existence of a stability surrogate from $z$). *There exists a measurable function* $r : \mathbb{R}^k \to [0, 1]$ *such that*

$$\mathbb{E}[|r(z) - \alpha|] \leq 1 - \eta_\phi. \tag{15}$$

*Proof.* Fix an anchor $z$. Define a randomized agreement predictor $\hat{A}$ for a counterfactual score $S = S(\delta)$ as follows. Independently sample a reference score $R$ from the *opposite* conditional distribution given $z$:

$$R \sim \begin{cases} S^- \mid z, & \text{if } A = 1, \\ S^+ \mid z, & \text{if } A = 0, \end{cases}$$

and output $\hat{A} := \mathbf{1}\{S \geq R\}$. This construction is used only to relate agreement separation to an achievable agreement-inference error.

Condition on $z$. If $A = 1$, then $S \sim S^+ \mid z$ and $R \sim S^- \mid z$ independently, and the error event is $\{S < R\}$:

$$\Pr(\hat{A} \neq A \mid A = 1, z) = \Pr(S^+ < S^- \mid z) = 1 - \eta_\phi(z).$$

If $A = 0$, then $S \sim S^- \mid z$ and $R \sim S^+ \mid z$ independently, and the error event is $\{S \geq R\}$:

$$\Pr(\hat{A} \neq A \mid A = 0, z) = \Pr(S^- \geq S^+ \mid z) \leq 1 - \Pr(S^+ > S^- \mid z) \leq 1 - \eta_\phi(z).$$

Therefore, for all $z$,

$$\Pr(\hat{A} \neq A \mid z) \leq 1 - \eta_\phi(z), \qquad \text{and hence} \qquad \Pr(\hat{A} \neq A) \leq 1 - \eta_\phi. \tag{16}$$

Now define $r(z) := \Pr_{\delta \sim \mathcal{D}}[\hat{A} = 1 \mid z]$. For a fixed example $(x, r, a)$, let $p := \Pr_\delta(A = 1)$ and $\hat{p} := \Pr_\delta(\hat{A} = 1)$. By the standard coupling inequality for Bernoulli variables,

$$|\hat{p} - p| = |\mathbb{E}_\delta[\hat{A} - A]| \leq \mathbb{E}_\delta[|\hat{A} - A|] = \Pr_\delta(\hat{A} \neq A).$$

Taking expectation over $(x, r, a, y) \sim \mathcal{P}$ yields $\mathbb{E}[|r(z) - \alpha|] \leq \Pr(\hat{A} \neq A) \leq 1 - \eta_\phi$, proving (15). $\square$

### N.4. From stability to hallucination detection with supervised probing

Define the oracle benchmark using the true stability score:

$$e_\alpha := \inf_T \Pr(\mathbf{1}\{\alpha \geq T\} \neq y). \tag{17}$$

**Assumption N.3** (Threshold regularity of $\alpha$). Let $T^\star \in \arg\min_T \Pr(\mathbf{1}\{\alpha \geq T\} \neq y)$ be an optimal threshold. There exists $B > 0$ such that for all $\epsilon \in [0, 1]$,

$$\Pr(|\alpha - T^\star| \leq \epsilon) \leq B\epsilon. \tag{18}$$

**Lemma N.4** (Plug-in thresholding with an approximate stability surrogate). *Under Assumption N.3, for any score $\tilde{\alpha} \in [0, 1]$, the detector $\hat{y} = \mathbf{1}\{\tilde{\alpha} \geq T^\star\}$ satisfies*

$$\Pr(\hat{y} \neq y) \leq e_\alpha + B \, \mathbb{E}\big[|\tilde{\alpha} - \alpha|\big]. \tag{19}$$

*Proof.* Let $\hat{y}^\star = \mathbf{1}\{\alpha \geq T^\star\}$. Then

$$\Pr(\hat{y} \neq y) \leq \Pr(\hat{y}^\star \neq y) + \Pr(\hat{y} \neq \hat{y}^\star) = e_\alpha + \Pr(\mathbf{1}\{\tilde{\alpha} \geq T^\star\} \neq \mathbf{1}\{\alpha \geq T^\star\}).$$

The disagreement event implies $|\alpha - T^\star| \leq |\tilde{\alpha} - \alpha|$. Condition on $|\tilde{\alpha} - \alpha|$ and apply Assumption N.3 to obtain $\Pr(\hat{y} \neq \hat{y}^\star) \leq B \, \mathbb{E}[|\tilde{\alpha} - \alpha|]$. $\square$

### N.5. Proof of Proposition 4.2

To relate the existence of the thresholding rule $\mathbf{1}\{r(z) \geq T^\star\}$ to a *supervised probe*, an explicit approximation term is included below.

**Assumption N.5** (Probe approximation). Let $\mathcal{H}$ be the hypothesis class used for supervised probing (e.g., linear classifiers or a small MLP). There exists $h \in \mathcal{H}$ such that

$$\Pr(h(z) \neq \mathbf{1}\{r(z) \geq T^\star\}) \leq \epsilon_{\text{probe}}. \tag{20}$$

**Proposition N.6** (Agreement separation $\Rightarrow$ bounded error of supervised probing). *Assume Assumptions N.3–N.5. Let $\eta_\phi$ be defined in Definition N.1. There exists a supervised probe $h$ on shaped embeddings $z$ such that*

$$\Pr(h(z) \neq y) \;\leq\; e_\alpha \;+\; B\,(1 - \eta_\phi) \;+\; \epsilon_{\text{probe}}. \tag{21}$$

*Proof.* By Lemma N.2, there exists $r : \mathbb{R}^k \to [0,1]$ such that $\mathbb{E}[|r(z) - \alpha|] \leq 1 - \eta_\phi$. Apply Lemma N.4 with $\tilde{\alpha} = r(z)$: the classifier $\bar{y} = \mathbf{1}\{r(z) \geq T^\star\}$ satisfies

$$\Pr(\bar{y} \neq y) \leq e_\alpha + B\,\mathbb{E}[|r(z) - \alpha|] \leq e_\alpha + B\,(1 - \eta_\phi).$$

Finally, by Assumption N.5 and a union bound,

$$\Pr(h(z) \neq y) \leq \Pr(\bar{y} \neq y) + \Pr(h(z) \neq \bar{y}) \leq e_\alpha + B\,(1 - \eta_\phi) + \epsilon_{\text{probe}}.$$

$\square$

**Discussion.** The bound decomposes hallucination detection error into: (i) $e_\alpha$, an oracle benchmark reflecting how predictive stability $\alpha$ is of truthfulness on the task; (ii) a label-free term $(1 - \eta_\phi)$ directly promoted by the shaping objective in Eq. 5; and (iii) a probe approximation term $\epsilon_{\text{probe}}$ that depends on the chosen probing class. This aligns with the method: ARS shapes embeddings to increase agreement separation, and the downstream detector is a supervised probe trained directly on $(z, y)$.

