# OpenReview forum: "Harnessing Reasoning Trajectories for Hallucination Detection via Answer-agreement Representation Shaping"
_ICML.cc/2026/Conference — ICML 2026 regular_

### Official Review · Reviewer_8bC8 · 2026-02-28

**Soundness:** 3
**Presentation:** 3
**Significance:** 3
**Originality:** 3
**Overall Recommendation:** 4
**Confidence:** 4

**Summary:**

This paper addresses the problem of hallucination detection in large reasoning models (LRMs), which may produce fluent reasoning trajectories while arriving at incorrect answers. The authors argue that directly using reasoning traces for detection is unreliable because surface-level trajectory patterns often fail to reflect answer validity. To overcome this limitation, they propose Answer-agreement Representation Shaping (ARS), a method that learns detection-oriented representations by explicitly encoding answer stability. ARS generates counterfactual answers through small latent perturbations of trajectory embeddings and labels these perturbations based on whether the resulting answers agree with the original output. The model is then trained to cluster answer-agreeing states and separate answer-disagreeing states, thereby exposing latent instability signals associated with hallucination risk. The shaped embeddings are designed to be plug-and-play with existing embedding-based detectors and require no additional human annotations. Experimental results across multiple benchmarks show that ARS consistently improves hallucination detection performance over strong baselines.

**Compliance With Llm Reviewing Policy:**

Affirmed.

**Key Questions For Authors:**

Key Questions for Authors

1. Sensitivity to Perturbation Design:
How sensitive is ARS to the magnitude, distribution, and number of latent perturbations applied to trajectory embeddings? Have the authors evaluated stability across different perturbation scales or alternative intervention strategies? A robustness analysis here could strengthen the soundness of the approach.

2.Theoretical Justification of Instability Signal:
The method assumes that small latent interventions exposing answer disagreement correlate with hallucination risk. Can the authors provide either theoretical intuition or empirical evidence (e.g., controlled experiments) that justify this assumption beyond observed performance gains?

3. Computational Overhead and Scalability:
What is the computational cost of generating counterfactual perturbations during training, and how does this scale with larger reasoning models or longer trajectories? Is ARS practical for frontier-scale LRMs?

4. Comparison to Self-Consistency and Confidence-Based Methods:
How does ARS differ fundamentally from self-consistency-based detection or confidence/entropy-based uncertainty measures? A more explicit comparative analysis (theoretical or empirical) would clarify its distinct contribution.

5. Failure Case Analysis:
Can the authors provide qualitative examples where ARS succeeds and fails? Understanding scenarios where representation shaping does not capture hallucination risk would help assess generalization.

**Limitations:**

The authors discuss some practical considerations of their approach, but the limitations section could be strengthened. In particular, the dependence on latent perturbation design and the potential sensitivity to hyperparameters are not fully explored. The scalability of ARS to very large reasoning models and longer reasoning trajectories remains somewhat unclear. Additionally, while the method aims to improve hallucination detection, it does not directly address underlying generation errors, and false positives in detection could affect downstream system behavior. From a societal perspective, improved hallucination detection may enhance model reliability, but deployment decisions and threshold settings could influence user trust and over-reliance on automated systems. A more explicit discussion of these aspects would improve transparency.

**Strengths And Weaknesses:**

Strengths

Soundness:
The paper presents a technically coherent framework for hallucination detection based on representation shaping. The key idea—using latent perturbations of reasoning trajectory embeddings to induce answer agreement or disagreement signals—is clearly motivated and methodologically well defined. The perturbation-and-labeling strategy provides a principled way to extract answer stability information without requiring additional human annotations. Experimental evaluation appears systematic, with comparisons against strong baselines and ablation studies demonstrating the contribution of the shaping mechanism. The plug-and-play design also strengthens methodological credibility, as it integrates with existing embedding-based detectors rather than relying on specialized architectures.

Presentation:
The paper is generally well structured, with a clear progression from problem motivation to methodological formulation and experimental validation. The conceptual framing around “answer stability” and “latent instability” provides an intuitive narrative that helps explain the approach. Figures and methodological descriptions aid understanding of the perturbation and shaping pipeline. The distinction between trajectory surface features and latent answer-consistency signals is clearly articulated.

Significance:
Hallucination detection in large reasoning models is a timely and important problem, especially as reasoning-augmented LLMs become widely deployed. The paper targets a core limitation of trajectory-based detection methods and proposes a mechanism that could be broadly applicable to different LRM architectures. If robust, the proposed shaping approach may influence future work on internal representation analysis, stability-based uncertainty estimation, and self-consistency modeling in reasoning systems.

Originality:
While the work builds on existing ideas in representation learning and perturbation-based analysis, it introduces a novel combination: explicitly shaping reasoning trajectory embeddings according to answer agreement signals induced by latent interventions. The notion of training embeddings to cluster answer-consistent states and separate unstable ones offers a fresh perspective on hallucination detection that differs from confidence-based or output-probability approaches. The contribution lies more in the conceptual reframing and representation-level intervention than in introducing a new architecture.

Weaknesses

Soundness:
Although the perturbation-based labeling mechanism is intuitive, the theoretical grounding for why small latent interventions reliably expose hallucination risk could be further strengthened. It remains somewhat unclear how sensitive the method is to the magnitude and distribution of perturbations, and whether performance depends heavily on hyperparameter tuning. Additionally, while empirical gains are demonstrated, more analysis on failure cases and robustness across diverse reasoning domains would strengthen confidence in generalization.

Presentation:
Some aspects of the method could benefit from clearer formalization, particularly the exact optimization objective and how representation shaping interacts with downstream detectors. A more explicit comparison to related stability-based or self-consistency detection methods would help situate the contribution more precisely within existing literature.

Significance:
The improvements, while consistent, may be incremental relative to strong baselines, depending on the scale of gains reported. It is not yet fully demonstrated whether ARS scales effectively to frontier models or highly diverse real-world tasks. The practical deployment implications (e.g., computational overhead of perturbation generation) could also be discussed more thoroughly.

Originality:
The approach relies on representation learning and perturbation strategies that have conceptual precedents in contrastive learning and uncertainty estimation. While the specific formulation is novel, some reviewers may view the contribution as an adaptation and recombination of existing ideas rather than a fundamentally new paradigm.

---

> ### Author Rebuttal · Authors · 2026-03-29
>
> We thank the reviewer for the detailed and thoughtful feedback, and address your questions below.
>
> **A1. Sensitivity to perturbation design**
>
> We agree that robustness to perturbation design is important. The current paper already provides several analyses. **Figure 3b** shows that performance improves with moderate perturbation strength $\sigma$ and declines when perturbations are too weak or too strong. **Figure 3c-d** further shows that ARS is stable across a broad range of shaping-head dimensions $k$ and counterfactual sample numbers $M$, with gains saturating at moderate values.
>
> Further, **Table 4** compares ARS with deletion, masking, and paraphrasing, showing that latent trace-boundary intervention performs best. This suggests ARS benefits from perturbations that preserve answer relevance while revealing nearby instability, rather than arbitrary text corruption. We also tested alternative perturbation distributions on TruthfulQA with Qwen3-8B:
>
> | Method | Ours | Uniform sphere | Uniform [-1.75, 1.75] |
> |---|---:|---:|---:|
> | ARS (CCS) | 86.64 | 82.12 | 82.54 |
> | ARS (Probing) | 83.66 | 83.71 | 83.06 |
>
> These results show that our design is more effective overall.
>
> **A2. Scalabity to larger models and broader reasoning domains**
>
> We additionally evaluate ARS on the larger Qwen3-32B model on TruthfulQA, where the gains remain substantial:
>
> | Method | Qwen3-32B |
> |---|---:|
> | CCS | 51.89 |
> | ARS (CCS) | 75.96 |
> | Supervised Probing | 69.78 |
> | ARS (Probing) | **77.85** |
>
> We also test a different reasoning domain with longer reasoning traces: the `causal_judgment` subset of Big-Bench Hard. Results on Qwen3-8B further support generalizability:
>
> | Method | BBH |
> |---|---:|
> | CCS | 65.30 |
> | ARS (CCS) | 79.50 |
> | Supervised Probing | 60.62 |
> | ARS (Probing) | **83.88** |
>
> > [1] Mirac Suzgun et al., *Challenging BIG-Bench Tasks and Whether Chain-of-Thought Can Solve Them*, arXiv preprint, 2022.
>
> **A3. Why should local instability correlate with hallucination risk?**
>
> The paper provides both theoretical and empirical support. Theoretically, **Proposition 4.2** bounds hallucination detection error by terms involving agreement separation and the informativeness of answer stability for truthfulness. Empirically, **Lines 414–422** show that counterfactual consistency alone already gives 80.7% thresholding accuracy on GSM8K with Qwen3-8B. Intuitively, truthful answers should occupy a larger local latent region that preserves the same answer, whereas hallucinated answers are more fragile under small perturbations.
>
> **A4. Computational overhead**
>
> ARS is lightweight at deployment: the base LRM is frozen, only a small shaping head is trained, and perturbations are used only during training. No extra sampling is needed at test time, unlike self-consistency methods. As reported in **Appendix F**, ARS training on TruthfulQA with Qwen3-8B takes 649 seconds. At inference, ARS+CCS takes 0.0194 seconds on the test set, compared with 1032 seconds for Semantic Entropy and 0.2889 seconds for G-Detector, while also achieving stronger detection performance.
>
> **A5. Consistency and confidence-based methods**
>
> ARS is fundamentally different from both methods. Self-consistency methods aggregate multiple sampled outputs at inference time, while confidence-based methods rely on logits, entropy, or verbalized confidence. In contrast, ARS uses latent perturbations only during training to create answer-agreement supervision, then learns a representation that can be scored from a single generation at test time. This makes ARS both plug-and-play and inference-efficient. **Tables 1 and 2** already compare against logit-based, self-consistency-based, verbalized-confidence, and recent LRM-specific baselines, with ARS achieving the strongest overall results.
>
> **A6. Failure cases**
>
> We agree that more discussion of failure cases would strengthen the paper. The current submission already includes qualitative evidence in **Figure 5**, which shows that ARS yields more separable scores and more discriminative embeddings than vanilla LRM representations. In the revision, we will further discuss representative failure modes, such as incorrect but locally stable answers and noise in answer-equivalence judgments.
>
>
> **A7. Presentation**
>
> Thank you for this suggestion. We placed the detailed optimization process in **Appendix I** and will consider moving some of it to the main paper for clarity. We also appreciate the suggestion to discuss limitations more explicitly and will incorporate this into the revised paper and impact statement.

---

### Official Review · Reviewer_Cgms · 2026-03-12

**Soundness:** 3
**Presentation:** 3
**Significance:** 2
**Originality:** 3
**Overall Recommendation:** 4
**Confidence:** 3

**Summary:**

This paper introduces Answer-agreement Representation Shaping (ARS), a framework for improving hallucination detection in large reasoning models (LRMs). ARS generates counterfactual answers by perturbing the last token embedding of the reasoning trajectory and creates new answers based on small latent interventions. The method then shapes the answer representations by comparing whether the counterfactual answers agree with the original answer. By clustering agreeing answers together and pushing disagreeing answers apart, ARS optimizes answer embeddings to reflect answer stability, avoiding brittle text perturbations. Experiments demonstrate that ARS significantly improves detection performance, achieving state-of-the-art results on multiple benchmarks.

**Compliance With Llm Reviewing Policy:**

Affirmed.

**Final Justification:**

Thanks for the clarification during the rebuttal period. I would like to maintain my current given score.

**Key Questions For Authors:**

1. In the Strengths and Weaknesses section, it was mentioned that the intervention focuses on the last token. Could you please clarify why this choice was made, and whether experimenting with the last 1% or 2% of tokens might offer additional insights?
2. In addition to AUROC, were other evaluation metrics considered in your experiments? If so, did those metrics also yield state-of-the-art (SOTA) results?

**Limitations:**

1. The method has not been tested on a broader range of models with different sizes.
2. The intervention focuses solely on the last token, without considering alternative intervention strategies based on percentage of tokens, such as the last 1% or 2%.

**Strengths And Weaknesses:**

Strengths:

1. The paper is technically sound, with claims that are well-supported by empirical experiments. The experiments are well-designed and adequately address the research problem. The paper is clearly written and well-structured, with an adequate appendix to support the main content. The authors address an important and relevant problem in hallucination detection for large reasoning models (LRMs).
2. The paper demonstrates excellent originality by combining existing techniques with a novel perspective. It makes a significant contribution by focusing on answer-agreement shaping and latent-state perturbations, proposing a new method that uses counterfactual answers and their agreement with the original answers to shape model embeddings.

Weaknesses:

1. While ARS shows promising results with 8B and 14B models, it would strengthen the paper if the method's performance were also demonstrated on smaller models (less than 8B) and larger models (greater than 14B). Adding 1-2 additional models would help further demonstrate the efficiency and scalability of ARS across different model sizes.
2. The choice of intervention position. The authors find that the intervention at the trace boundary performs the best, but why focus solely on the last token instead of testing the last 1% or 2% of tokens? Testing these could make the results more convincing, as the effect of the final token differs in longer and shorter reasoning chains. For longer reasoning, the final token's effect might be smaller, while for shorter reasoning, it could play a more significant role.

---

> ### Author Rebuttal · Authors · 2026-03-29
>
> We thank the reviewer for the constructive feedback and for recognizing the originality and technical soundness of ARS. We are glad that the reviewer found the paper well structured and the contribution meaningful for hallucination detection in LRMs.
>
> **A1. Broader model coverage**
>
> We agree that broader scale coverage would further strengthen the paper. We report additional experiments on Qwen3-4B and 32B models on TruthfulQA dataset as follows, where the effectiveness of our approach still holds.
>
> |     Method   | Qwen3-4B | Qwen3-32B|
> | ------ | ----- | ----- |
> |CCS |58.98 | 51.89 |
> |ARS (CCS) |**83.90** | 75.96|
> |Supervised Probing |75.29 | 69.78|
> |ARS (Probing) | 83.64|**77.85**|
>
> **A2. The choice of intervention position**
>
> Thank you for your suggestion. As suggested, we intervene at the last 1% tokens on TruthfulQA dataset and Qwen3-8B model. The results show that intervening at the last several tokens is similarly effective while slightly worse than our choice. We will add these ablation results to the final version.
>
> |     Method   | Last token (Ours) | Last 1% tokens| Last 2% tokens|
> | ------ | ----- | ----- | ----- |
> |ARS (CCS) | **86.64**| 81.70 |83.32|
> |ARS (Probing) | **83.66** | 82.16 |82.83|
>
>
> **A3. Metrics beyond AUROC**
>
> We report the Accuracy, F1 and AUPRC scores on TruthfulQA dataset and Qwen3-8B model, and ARS can still maintain a considerable improvement over detection on the vanilla embedding space, and a competitive baseline TSV.
>
>
> |     Method   | F1 | AUPRC|Accuracy |
> | ------ | ----- | ----- | ----- |
> | TSV | 64.03| 68.01 |74.63|
> |CCS       | 69.35| 77.61 |55.80|
> |ARS (CCS) |**74.67** |**87.60** |66.47|
> |Supervised Probing |57.60 | 55.29|67.80|
> |ARS (Probing) |71.95 |75.15|**77.56**|

---

> > ### Author Rebuttal · Reviewer_Cgms · 2026-04-03
> >
> > Thanks for your feedback, and I have no more questions for this paper. I believe I already gave a considerate score for this paper. Good luck!

---

> > > ### Author Response · Authors · 2026-04-03
> > >
> > > Thank you for taking time to read our rebuttal and we are glad that your concerns have been addressed!
> > >
> > > Best,
> > >
> > > Authors

---

### Official Review · Reviewer_6YGK · 2026-03-13

**Soundness:** 3
**Presentation:** 4
**Significance:** 3
**Originality:** 3
**Overall Recommendation:** 5
**Confidence:** 4

**Summary:**

This work studies studies hallucination detection for large reasoning models (LRMs) by leveraging internal representations around the reasoning–answer boundary. They introduce Answer-agreement Represent Shaping (ARS), which can be used to detect if the model is hallucinating by leveraging the hidden state at the reasoning boundary (right before answer token and final layer). In ARS, they perturb the hidden states and see if the decoded answers change and follow the intuition that hallucinated answers often correspond to fragile internal states, while truthful answers are supported by more stable representations.

Using these perturbations, the method constructs answer-agreement and answer-disagreement embedding pairs, which are then used to train a lightweight mapping that transforms the vanilla answer representation into a shaped embedding that better reflects answer stability. At inference time, the shaped embedding can be directly fed into existing embedding-based hallucination detection scoring functions.

The results show that the shaped representations show substantial improvement over vanilla embeddings, and surpass various baseline methods. They also show that the learned mapping transfers reasonably well across datasets, suggesting that the stability signal captured by ARS generalizes beyond the training distribution.

**Compliance With Llm Reviewing Policy:**

Affirmed.

**Final Justification:**

Thanks for the response and thoughtful discussion. I am happy to maintain my score.

**Key Questions For Authors:**

1. Does the framework consider cases where the model abstains or expresses uncertainty explicitly? How do responses with abstention fit into the representation shaping?

**Limitations:**

yes

**Strengths And Weaknesses:**

### Soundness
The proposed method is technically reasonable and well motivated. The core hypothesis—that truthful answers correspond to more stable internal states under small latent perturbations—is intuitive and the implementation is simple and clean, requiring only a lightweight mapping. The experimental setup is generally sound and compares against a wide range of baselines across multiple datasets.

### Presentation
The paper is well written and easy to follow. The figures and tables are well designed and effectively support the presentation of the method and experimental results.

### Significance
This paper studies the timely and interesting problem of hallucination in large reasoning models and propose to detect hallucination based on the reasoning traces. The authors demonstrates that the representation shaping proposed in this work is light-weight and can be applied to various embedding-based hallucination detection. The generalization across datasets might indicate potential usages and insights beyond hallucination detection.

### Originality
The paper proposes an elegant idea that combines latent perturbations with contrastive representation shaping to expose answer instability signals. While related ideas exist in consistency-based uncertainty estimation and representation-based hallucination detection, the specific formulation of using latent perturbations at the reasoning boundary to construct agreement/disagreement supervision appears novel.

---

> ### Author Rebuttal · Authors · 2026-03-29
>
> We sincerely thank the reviewer for the positive evaluation. We are especially encouraged that you found the method intuitive, lightweight, and well motivated, and that you highlighted its potential generalization beyond the training distribution.
>
> **A1. How does ARS handle abstentions or explicit uncertainty?**
>
> This is an excellent question. In the current formulation, ARS treats abstentions as ordinary answers under the same answer-agreement function $\rm Agr(a,\tilde{a})$: if the original answer and a counterfactual answer are judged equivalent abstentions, they naturally form an agreement pair; if one abstains while another gives a contentful or conflicting answer, they form a disagreement pair. In this sense, abstention is already compatible with the current framework through the same agreement-based pipeline. That said, we did not isolate abstention as a separate evaluation regime in the current submission, and we agree that making this scope explicit would improve the paper.
>
> A natural abstention-aware evaluation would separate responses into *answer* vs. *abstain* cases, then measure whether ARS assigns lower hallucination risk to correct abstentions than to incorrect substantive answers, and whether shaping remains effective within each subset. We will clarify this in the revision and add discussion on abstention-aware evaluation as an important future direction.

---

> > ### Author Rebuttal · Reviewer_6YGK · 2026-04-03
> >
> > Thanks for the response and thoughtful discussion. I am happy to maintain my score.

---

> > > ### Author Response · Authors · 2026-04-03
> > >
> > > Thank you for taking the time to read our rebuttal and maintain your positive score!
> > >
> > > Best,
> > >
> > > Authors

---

### Decision · Program_Chairs · 2026-04-30

**Decision:**

Accept (regular)

**Comment:**

The paper introduces Answer-agreement Representation Shaping (ARS), a framework that improves hallucination detection in large reasoning models by leveraging latent stability at the reasoning-answer boundary. The key idea of this paper is that truthful answers correspond to more stable internal representations, while hallucinations are more fragile under small perturbations. By using these perturbations as a supervisory signal during training, ARS learns to shape embeddings into a form that better distinguishes between correct and incorrect outputs.
The reviewers found the work technically sound and highly original. A significant strength is its efficiency. because latent interventions are only used during training, the method remains lightweight and much faster at inference time than sampling-based approaches like self-consistency.

During the rebuttal and discussion period, the authors successfully addressed concerns regarding scalability and robustness. They provided new experiments demonstrating that ARS generalizes well across different model sizes and reasoning domains, such as Big-Bench Hard. They also clarified that the method is robust to different perturbation scales and intervention positions. Given the strong empirical results, the practical efficiency of the plug-and-play design, and the successful resolution of all reviewer queries, I recommend this paper for acceptance.